# Deciphering ovarian cancer heterogeneity through spatial transcriptomics, single-cell profiling, and copy number variations

Songyun Li[1,2], Zhuo Wang[1,2]*, Hsien-Da Huang[1,2]*

**1** Warshel Institute for Computational Biology, The Chinese University of Hong Kong, Shenzhen, Guangdong, P.R. China, **2** School of Medicine, The Chinese University of Hong Kong, Shenzhen, Guangdong, P. R. China

* wangzhuo@cuhk.edu.cn (ZW); huanghsienda@cuhk.edu.cn (HDH)

## Abstract

High-grade serous ovarian carcinoma (HGSOC) poses a formidable clinical challenge due to multidrug resistance (MDR) caused by tumor heterogeneity. To elucidate the intricate mechanisms underlying HGSOC heterogeneity, we conducted a comprehensive analysis of five single-cell transcriptomes and eight spatial transcriptomes derived from eight HGSOC patients. This study provides a comprehensive view of tumor heterogeneity across the spectrum of gene expression, copy number variation (CNV), and single-cell profiles. Our CNV analysis revealed intratumor heterogeneity by identifying distinct tumor clones, illuminating their evolutionary trajectories and spatial relationships. We further explored the homogeneity and heterogeneity of CNV across tumors to pinpoint the origin of heterogeneity. At the cellular level, single-cell RNA sequencing (scRNA seq) analysis identified three meta-programs that delineate the functional profile of tumor cells. The communication networks between tumor cell clusters exhibited unique patterns associated with the meta-programs governing these clusters. Notably, the ligand-receptor pair MDK - NCL emerged as a highly enriched interaction in tumor cell communication. To probe the functional significance of this interaction, we induced NCL overexpression in the SOVK3 cell line and observed enhanced tumor cell proliferation. These findings indicate that the MDK - NCL interaction plays a crucial role in promoting HGSOC tumor growth and may represent a promising therapeutic target. In conclusion, this study comprehensively unravels the multifaceted nature of HGSOC heterogeneity, providing potential therapeutic strategies for this challenging malignancy.

## Introduction

High-grade serous ovarian carcinoma(HGSOC), characterized by its aggressive behavior and rapid proliferation, is the most prevalent and lethal form of ovarian cancer, accounting for over 70% of ovarian cancer-related fatalities [1]. One of the primary challenges in treating high-grade serous ovarian cancer (HGSOC) is multidrug resistance (MDR), where cancer cells develop resistance to various chemotherapy agents, particularly platinum-based and taxane therapies, resulting in treatment failure [2]. Platinum-derived drugs generate reactive

**Data availability statement:** The datasets analyzed during the current study are available in the Gene Expression Omnibus (GEO) repository with the identifier(s) GSE211956 [14]. Code for all the analysis can be found in https://github.com/SongyunLi21/Ovarian-cancer-heterogeneity

**Funding:** This work is supported by the Warshel Institute for Computational Biology funding from Shenzhen City and Longgang District (LGKCSDPT2024001); Natural Science Foundation of Guangdong(2023A1515011861); National Natural Science Foundation of China (No.32070674); Shenzhen Science and Technology Program(JCYJ20220530143615035). There was no additional external funding received for this study. The funders had no role in study design, data collection and analysis, decision to publish, or preparation of the manuscript.

**Competing interests:** The authors have declared that no competing interests exist.

oxygen species (ROS), leading to DNA damage and apoptosis. However, HGSOC cells bolster their antioxidant defenses by upregulating antioxidants such as glutathione and NAD(P)H: quinone oxidoreductase 1, which contributes to their chemoresistance [3,4]. MDR is caused by the inconsistent response of tumor cells to treatment, a variability rooted in the diverse and heterogeneous nature of tumor cells [5,6]. Tumor heterogeneity refers to the diversity and variability of cells within a single tumor or among different tumors of the same type [7]. This diversity plays a crucial role in tumor progression and influences the response to treatment.

DNA microarrays and next-generation sequencing (NGS) have become indispensable tools for deciphering intratumor heterogeneity. These technologies have been instrumental in characterizing heterogeneity in single nucleotide polymorphisms (SNPs), tracking the evolution of heterogeneity [8], elucidating the connection between metastasis and CNVs[9], and establishing associations between intratumor heterogeneity and clinical outcome [10]. Recently, the cellular heterogeneity of HGSOC has been revealed by scRNA seq, including the heterogeneity of tumor cells [11] and tumor-infiltrated immune cells [12–15]. However, despite these breakthroughs, the spatial and histological context of the tumor remains elusive, resulting in a lack of information about the spatial localization of cellular components and molecules. Spatial transcriptomics (ST) is emerging as a transformative approach to address this gap. By sequencing RNA from spatially defined regions on a tissue section, this method enables researchers to map gene expression levels onto the tissue, offering a comprehensive understanding of spatial heterogeneity.

Genetic alterations play a pivotal role in tumor development, which leads to abnormal expression of oncogenes, tumor suppressors, and the activation of tumor-related pathways. Copy number variations(CNVs) is a phenomenon in which sections of the genome are repeated and the number of repeats in the genome varies between individuals, which are instrumental in identifying the origins of tumors. The process of clonal evolution, characterized by the accumulation of somatic genetic changes during tumor development, underlies tumor heterogeneity.

According to the principles of Darwinian evolution, clonal evolution begins with a single cell acquiring an initial genetic mutation that promotes uncontrolled proliferation. As the tumor progresses, additional genetic mutations accumulate randomly, affecting various genes, including oncogenes and tumor suppressors. Over time, persistent clonal evolution leads to a diverse array of genetic alterations, contributing to the observed heterogeneity in HGSOC, including variations in pathway activity and responses to treatment [16].

CNV analysis is essential for constructing a clonal evolution tree, facilitating the identification and tracking of different clones within tumors. While some previous research has used CNVs to explain aspects of tumor heterogeneity [17–19], there is a lack of studies examining both heterogeneity and the spatial relationships between clones. Our research focuses on intratumor heterogeneity among distinct clones, correlating these observations with CNVs. Moreover, utilizing ST data, we explore the spatial distribution of different clones, enhancing our understanding of heterogeneity development. Our study aims to contribute significantly to understanding the intricate dynamics of tumor evolution and its implications for personalized therapeutic strategies.

To offer a comprehensive understanding of HGSOC heterogeneity, we conducted an extensive analysis using data from five scRNA-seq experiments and eight ST datasets derived from eight distinct HGSOC cases. Our investigation took a multi-dimensional approach, exploring intratumor, intertumor, and cellular through gene expression, CNV, and single-cell profiles. To unravel the origins of this heterogeneity, we examined both the uniformity and diversity of CNV patterns across these tumors. Additionally, our study identified a significant enrichment of macrophages in HGSOC, with prognostic implications. We further analyzed the

communication between these macrophages and tumor cells, aiming to elucidate their role in tumor progression. Our study provides a comprehensive depiction of the spatial heterogeneity within HGSOC, providing valuable insights for the development of targeted and personalized clinical therapy.

## Materials and methods

### Data sources

Both the scRNA seq and ST data were obtained from the Gene Expression Omnibus (GEO), specifically from dataset GSE211956. The ST datasets contained data collected from eight patients who had undergone neoadjuvant chemotherapy involving taxane and platinum-based treatment. The scRNA seq data included samples from five of these patients post-neoadjuvant chemotherapy [20]. To enhance the reliability of our analysis, the scRNA seq data have been preprocessed and annotated to ensure the quality and accuracy of the information utilized. This annotated scRNA seq data was directly used in our study.

### ST data processing

We initially conducted quality control on the dataset from Space Ranger using Seurat 4.3.0 [21]. To exclude regions of damaged tissue, spatial spots with fewer than 400 genes in the ST datasets obtained from GEO were omitted. In our proprietary datasets, spatial spots were required to have less than 13% mitochondrial gene content and retain between 200 and 4,000 genes.

To ensure harmonized expression data across different patients, we employed the R package harmony (Version 0.1.1) [22]. We amalgamated the expression matrices from each patient's sections and performed a series of data preprocessing steps, including normalization, log transformation, centering, and scaling. The section source was designated as the batch factor, and the "RunHarmony" function was applied to mitigate batch-induced variability. Subsequently, the integrated datasets were analyzed using Seurat. The parameter "reduction = harmony" was applied, and 2000 highly variable genes were identified based on their expression means and variances.

Building upon a foundation of key genes, we carried out principal components analysis (PCA) to project the spatial spots into a lower-dimensional space defined by the first 20 principal components (PCs). This dimensionality reduction step facilitated the subsequent application of the corrected PC matrices in unsupervised shared-nearest-neighbor (SNN) analysis, clustering, and uniform manifold approximation and projection (UMAP) visualization, enabling us to discern distinct patterns within the dataset.

To identify differentially expressed genes within each cluster, we employed the FindAllMarkers() function. By comparing these marker genes to the established tumor and stromal signatures, we were able to accurately classify clusters into tumor, stromal, or an "unknown" category. To enhance our understanding of cluster distribution, we integrated sample names with cluster IDs, providing a more granular view of clusters within each sample. Subsequently, we carefully examined the distribution of each cluster across diverse samples. For further exploration of inter-tumor and stromal heterogeneity, we excluded clusters labeled as "unknown" and those with low distribution.

### Cell types related enrichment analysis

To assess the enrichment of cell type within the ST data, we conducted a multimodal intersection analysis (MIA) by integrating the scRNA-seq datasets with the ST datasets. The significance of the overlap between ST genes and cell type marker genes was determined using the

hypergeometric cumulative distribution, with all genes considered as the background for calculating the p - value. Simultaneously, we examined cell type depletion by computing -$\log_{10}^{(1 - p)}$ [23].

The scRNA-seq data, derived from differentially expressed genes (DEGs), had been previously annotated, enabling the characterization of DEGs for each cell type. We calculated DEGs within the retained tumor and stromal clusters. The extent of cell type enrichment was assessed using the MIA, taking into account the overlap between DEGs in the ST data and scRNA-seq data.

To visually present the enrichment of cell types for each spot, we utilized the ColSums() function to calculate the average expression of the annotated genes associated with each cell type. This calculation provided a representation of the enrichment within each spot.

### Analysis of clusters' similarity

For each cluster, the average gene expression was used as a representative value to perform principal component analysis (PCA). The first two PCs for each cluster were then used to enable visualization and subsequent hierarchical clustering [24].

### Analysis of signaling pathway

Using the gene set variation analysis (GSVA) package (Version 1.46.0) in R, we calculated a cluster-specific enrichment score by comparing the rank of expression values for genes within a reference gene set against the rank of expression values for all other genes. Specifically, we leveraged a set of 50 cancer hallmark signatures (MSigDB, H sets) and utilized the "gsva" function for analyzing pathway activity enrichment [25]. The raw expression matrix was utilized, with the average expression of each cluster as the input. From the clusters, we specifically selected pathways exhibiting the highest 50 or 30 variances across them. To visualize these results, hierarchical clustering was applied using Ward's minimum variance method.

### CNVs analysis and clonal tree construction

The infercnv package (Version 1.14.2) uses the "run" function for both tumor and unknown clusters with the following parameter settings: cutoff = 0.1, tumor_subcluster_partition_method = qnorm, denoise = T, HMM = F. This package utilizes gene expression patterns associated with specific chromosomal regions to infer CNVs based on gene expression [26–30]. The analysis was conducted for unknown and tumor clusters, using the stromal cluster as a reference. Similarly, in the absence of a reference, inferCNV was performed individually for each sample. The "infercnv.observations" file was then utilized to aggregate the CNV scores for each gene, providing a spatial representation of the CNV scores. This analysis was also extended to integrated datasets across all patients, using all the stromal clusters as a reference with cluster_by_groups = T. Hierarchical clustering was subsequently applied to these datasets.

To construct the clonal tree, CNVs within each sample were initially assessed using the same packages used in the previous CNV analysis. However, two additional parameter settings, specially HMM = T and cluster_by_groups = T, were utilized. Subsequently, the clonal tree was constructed using uphyloplot (Version 2.2.3) [31], taking "17 _HMM_predHMMi6.rand_trees.hmm_mode-subclusters.cell_groupings" as the input. The determination of CNVs within each clone was achieved by combining the "test_ groupings" file from clonal tree construction with the file obtained from inferCNV. The spatial location of each clone was then visualized by merging both the "test_groupings" and "17_HMM_predHMMi6.rand_trees.hmm_mode-subclusters.cell_groupings" files.

To investigate genes associated with CNVs, we leveraged the information contained in the "17_HMM_predHMMi6.rand_trees.hmm_mode-subclusters.genes_used.dat" file. By combining this data with the gene set of cancer hallmark signatures in GSVA, we established a link between the genes impacted by CNVs and the signaling pathways associated with these genes.

## Tumor score calculation and co-localization analysis

When computing the tumor score for each spot, a comprehensive approach was adopted by considering both CNV and tumor signatures. To calculate CNV scores, we first aggregated the CNV scores from all samples. Then, for a given spot denoted as "i", the CNV score was normalized using the follows formulation:

$$nCNV\_score_i = \frac{\left|nCNV\_score_i - 1\right|}{\left|max\left(CNV\_score\right) - 1\right|}$$

For the assessment of tumor signature scores, we used maximum-minimum normalization. The tumor score of a specific spot was calculated as the geometric mean of the normalized CNV score and the normalized tumor signature score.

To assess the co-localization of tumor cells with other cell types, we employed the chi-square test. Using the normalized cell signatures score, we categorized scores higher than the mean as "high", while the remaining scores were designated as "low". Notably, we identified cell types exhibiting at least eight p - p-values less than 0.05, indicating a statistically significant association, as indicative of co-localization with tumor cells.

## scRNA seq data processing and sub-clustering analysis

The scRNA-seq data obtained from GEO were preprocessed, eliminating the necessity for additional data manipulation. As tumor signatures sometimes fall short in identifying all tumor cells, we opted for manual cell-type annotation. The process was initiated with clustering and marker-based cell-type characterization. Epithelial and endothelial cells underwent CNV analysis, while other cells served as reference points. Following CNV characterization, we integrated dataset annotations with our CNV scores. Specifically, mesothelial and endothelial cells with normalized CNV scores exceeding 0.3 were designated as tumor cells, adhering to the initial annotation of tumor cells.

Next, we proceeded to distinguish between tumor cells and macrophages, employing PCA to streamline data by projecting spots based on the first 20 PCs. The resulting PC matrices were utilized for subsequent unsupervised shared-nearest-neighbor (SNN) clustering and UMAP visualization analysis. To further unravel the intricacies within each cluster, we identified DEGs within each cluster through the FindAllMarkers() function. Additionally, CellChat (Version 1.6.1) was used to unravel the cell-cell communication network between tumor cells and macrophages [32].

## Identify tumor meta-programs

To elucidate meta-programs within tumor cells, we applied non-negative factorization across all patients. This technique dissects the high-dimensional gene expression matrix into multiple lower-dimensional matrices featuring non-negative values. This resulting matrix unveils distinct data facets, facilitating the identification of underlying biological patterns. For each tumor, we determined the optimal number of meta-programs by applying the nmf() function with the "rank" parameter ranging from 2 to 6. The selection of the appropriate rank involved identifying the point just before a significant drop in the analysis, which served as our decisive criterion. Within each tumor

sample, we uncovered co-expressed gene modules using the nmf() function from the NMF package (Version 0.26). These modules were characterized by the top 50 genes with the highest weight, defining specific intratumor expression programs. To explore whether certain intratumor expression programs were shared across multiple tumors, we conducted a hierarchical clustering analysis on all programs. This analysis relied on the pairwise Jaccard index, calculated as the intersection of two intratumor programs (A and B) divided by their union [33].

$$\text{Jaccard index} = \frac{A \cap B}{A \cup B}$$

Intratumor programs shared by multiple tumors were designated as meta-programs, using genes shared by at least 1/3 of tumors within a specific meta-program for definition.

## Cell Culture

SKOV3 cells were cultured in McCoy's 5A Medium supplemented with penicillin, streptomycin, and 10% FBS(Fetal bovine serum). When the cells were 90% confluent, the old culture medium was discarded and the cells were rinsed twice with 2 mL PBS. Following the PBS wash, trypsin was added to the cell. The cells were monitored under a microscope for about 30 seconds. Once the cells exhibited a rounded morphology, 2 mL of complete culture medium was rapidly added to terminate the digestion. Cells were collected by gently pipetting, followed by centrifugation at 800 rpm, 4 °C for 5 minutes. The supernatant was carefully discarded. The cells were then resuspended in a complete culture medium and distributed in separate bottles. The cell culture medium was changed every other day.

The cells were divided into three groups. In the control group, cells were maintained without any transfection, serving as the baseline reference. The one-vector group was transfected with an empty plasmid, while the oe-NCL group was transiently transfected with the NCL expression plasmid.

## Plasmid and Cell Transfection

The full-length NCL gene was cloned into a pcDNA 3.1 (+) vector, incorporating ACC65I and EcoRI restriction sites. The primer sequences used for cloning NCL were: 5'-CGGGTACCATGGTGAAGCTCGCGAAGGC-3' and 5'-GGGAATTCCTATTCAAACTTCGTCTTCTTTCCTTGT-3'.

For transfection, cells were plated in a 24-well plate at approximately 70% confluency the day prior to the procedure and maintained under standard culture conditions. Transfections were carried out using 500 ng of plasmid DNA and 1.5 μL of Lipofectamine 3000 mixed with 150 μL of OPTI-MEM medium, incubated at room temperature for 5 minutes. This mixture was then combined with an additional 150 μL of OPTI-MEM containing the plasmid DNA and incubated for an additional 15 minutes in a sterile environment. Prior to transfection, the culture medium was removed from each well, and cells were rinsed three times with 1 mL of 1× PBS. Next, 400 μL of OPTI-MEM was added to each well, and the plates were returned to a 37°C incubator with 5% CO2. After 15 minutes, the transfection mixture (RNA and plasmid) was added according to experimental groupings, gently mixed, and incubated at 37°C for 6 hours. Following this incubation, the medium was replaced with a fresh serum-free medium. Transfection efficiency was assessed using PCR.

## Gene expression analysis with quantitative PCR

The total RNA was extracted from cells with a commercial kit. To obtain the cDNA, 2 μL of total RNA was used as a template, and the reverse transcription reaction system was set up following the kit instructions. The resulting cDNA was stored at -70 °C.

For the quantitative PCR (qPCR) reaction, the reaction system was prepared on an icebox, adhering to the procedures outlined in the reagent instructions. The premixed reagents were added to the 8-tube strip, and amplification was carried out in a PCR instrument. The acquired data were processed using the $2^{-\Delta\Delta Ct}$ formula, with the relative expression level of the target mRNA determined using the internal reference gene glyceraldehyde 3-phosphate dehydrogenase (*GAPDH*). The primer details are shown in Supplement S11.

## Western blot analysis

Cells were treated according to experimental groups, and total protein was extracted using RIPA buffer containing PMSF after cell lysis. The protein concentration was measured using the BCA protein assay. Equal amounts of protein were loaded onto SDS-PAGE gels (10% separation gel and 5% stacking gel) and separated by electrophoresis. The proteins were then transferred to PVDF membranes using a wet transfer method. The membranes were blocked with 5% non-fat milk in TBST and incubated with primary antibodies (e.g., NCL, PTEN, AKT, p-AKT, Ki67, GAPDH) overnight at 4°C. After washing, the membranes were incubated with HRP-conjugated secondary antibodies and developed using chemiluminescent substrates. Protein bands were visualized using a Tanon 5200 imaging system. Relative protein expression levels were quantified by analyzing band intensity using Image Pro Plus software. Statistical analysis was performed using one-way ANOVA, and significance was determined at $P < 0.05$.

## Cell proliferation analysis

Cells from each group were harvested through centrifugation and inoculated into a 96-well plate at a density of $1 \times 10^4$ cells per well, with each group comprising 12 wells. A culture medium was added according to the grouping, and cells were cultured for various durations. Cell proliferation was assessed using the CCK-8 kit, following the manufacturer's instructions. Signal detection was carried out using a microplate reader at a wavelength of 450 nm, and the cell proliferation of each group was calculated accordingly.

## Result

### Spatial and molecular heterogeneity in HGSOC

To gain a deeper understanding of the intricate spatial heterogeneity in HGSOC, we gathered ST data from eight HGSOC patients and scRNA-seq data from an additional five patients, sourced from the GEO dataset GSE211956. The number of spots of the ST data obtained from GEO included datasets ranging from 1,501 to 3,584 for each section, resulting in a total of 19,990 spots with a median of 2,459 genes and 5,882 unique molecular identifiers (UMIs) per spot following preprocessing.

To integrate the ST data from various patients and delineate the tumor and stromal regions, we conducted clustering and UMAP analysis, identifying 20 clusters (Fig 1B and S1 Table). Among these, five clusters (0, 2, 8, 13, 18) were annotated as tumor clusters due to their elevated expression of tumor markers (Figs 1B, 1D, and S1A), while an additional six clusters (1, 4, 5, 7, 15, 17) were identified as stromal clusters owing to their heightened expression of stromal markers (Figs 1B, 1D and S1A). The remaining clusters remained unclassified. To assess the malignancy of tumor clusters and unknown clusters, we used "infercnv" to characterize the CNVs in these clusters (Figs S2E and S2F). As anticipated, tumor clusters exhibited multiple CNVs and high malignancy. Interestingly, some subgroups across different clusters shared CNV patterns, and the spots within the same clusters did not necessarily exhibit identical CNVs. For unknown clusters, significant CNVs were observed, suggesting potential malignancy.

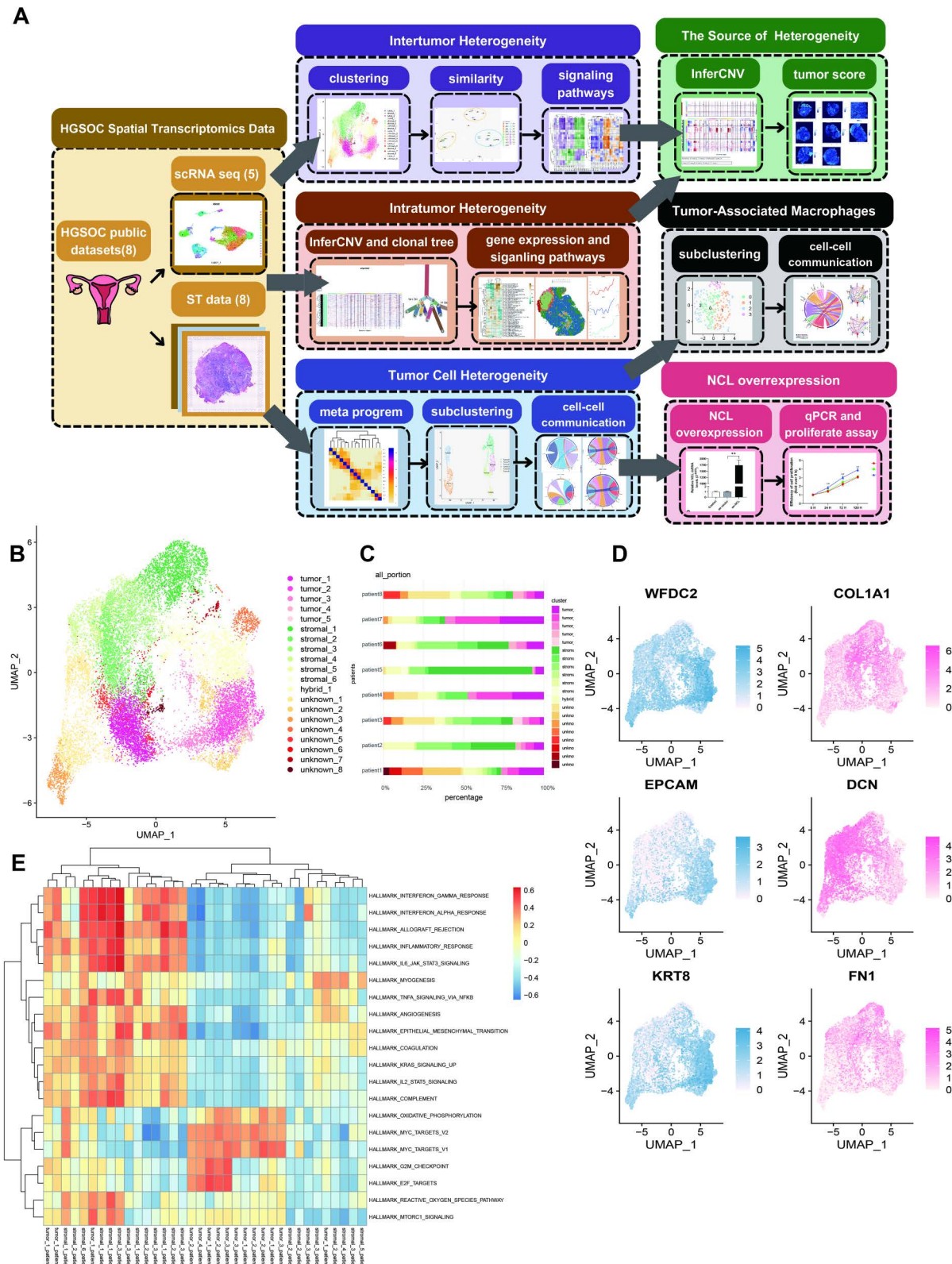

**Fig 1. Spatial transcriptomics analysis of 8 high grade serious ovarian cancer patients.** (A): Workflow of the study. (B): Uniform Manifold Approximation and Projection (UMAP) plot of clustering results of spatial transcriptomics (ST) data. (C): Cluster distributions in each patient. (D): UMAP plot of marker expression for tumor (blue) and stromal (pink) samples. (E): Heatmap of pathway activity in tumor and stromal clusters.

We then examined the distribution of clusters within each sample, revealing heterogeneity (Figs 1C, 2A, S1B, and S1C). Generally, all samples contained tumor_1 and tumor_2 clusters, which constituted a substantial portion of all tumor clusters. However, the presence and distribution of the other three clusters varied. For instance, the tumor_5 cluster was exclusively present in the samples of patient_2 and patient_3, with a predominant presence in patient_2, indicating heterogeneity in the distribution of tumor clusters. Similarly, stromal clusters displayed heterogeneity, with stromal_1, stromal_2, and stromal_3 making up a significant portion of stromal regions and being present in all samples. The remaining three clusters exhibited diverse distribution across samples, underscoring the complexity of spatial heterogeneity in both tumor and stromal regions within HGSOC.

To assess pathway activity, we focused on clusters predominantly present (comprising > 100 spots) in either the tumor or stromal regions within each sample and conducted GSVA (Fig 1E). In general, tumor regions exhibited significant enrichment in cellular proliferation pathways associated with the G1 phase, such as MYC targets, E2F targets, and G2 phase to mitotic phage (G2M) checkpoints, indicating uncontrolled cell growth within these areas. Conversely, the stromal region displayed notable enrichment in inflammatory responses, such as interleukin-signal transducer and activator of transcription (IL - STAT) response and interferon (INF) related response, and processes associated with tumor metastasis, including epithelial-mesenchymal transition and angiogenesis. Upon closer examination, these pathway enrichments displayed both intratumoral and intrastromal heterogeneity, underscoring their role in HGSOC heterogeneity. It is noteworthy that most stromal clusters in patient_8 did not exhibit these characteristics.

Through MIA, we confirmed the accuracy of our annotation for tumor and stromal regions. Tumor clusters showed high enrichment in mesothelial cells, tumor cells, and endothelial cells, while fibroblast cells were highly enriched in the stromal region (Fig S1D and S2 Table).

To assess gene expression similarities between different clusters, we conducted PCA and hierarchical clustering based on their gene expression profiles. Generally, clusters of the same type displayed shorter distances from each other (Fig 2B). However, tumor and stromal clusters were notably segregated into distinct groups. Consequently, we further investigated the heterogeneity within both tumor and stromal clusters, revealing intricate patterns that contribute to a comprehensive understanding of the spatial and molecular diversity within HGSOC.

## Tumor Heterogeneity Analysis through Gene Expression Clustering

Upon examining gene expression similarities between clusters using PCA and hierarchical clustering, a distinct division within the stromal clusters emerged. Notably, a higher degree of similarity was observed between stromal clusters from patient_2 and patient_5 (group 1) and clusters from patient_6 and patient_8 (group 2) (Fig 2D). The clusters from other patients exhibited characters resembling either of these two groups. This clustering pattern was evident not only in gene expression but also in hierarchical clustering based on the activity of signaling pathways. Specifically, these two groups were mainly characterized by whether they were highly enriched in pathways associated with tumor growth, inflammatory responses, and metastasis (group 1 being enriched, while group 2 was not).

To further investigate the differences between these two groups, we computed the average gene expression for each group and assessed cellular enrichment using MIA (Fig S2A) and xCell (Fig 2E). It is noteworthy that Group 2 exhibited a significantly higher enrichment of macrophages and T cells, yet both B cells and plasma cells were absent from the infiltration. This suggests a potential focus on inflammatory responses and tumor-specific T-cell

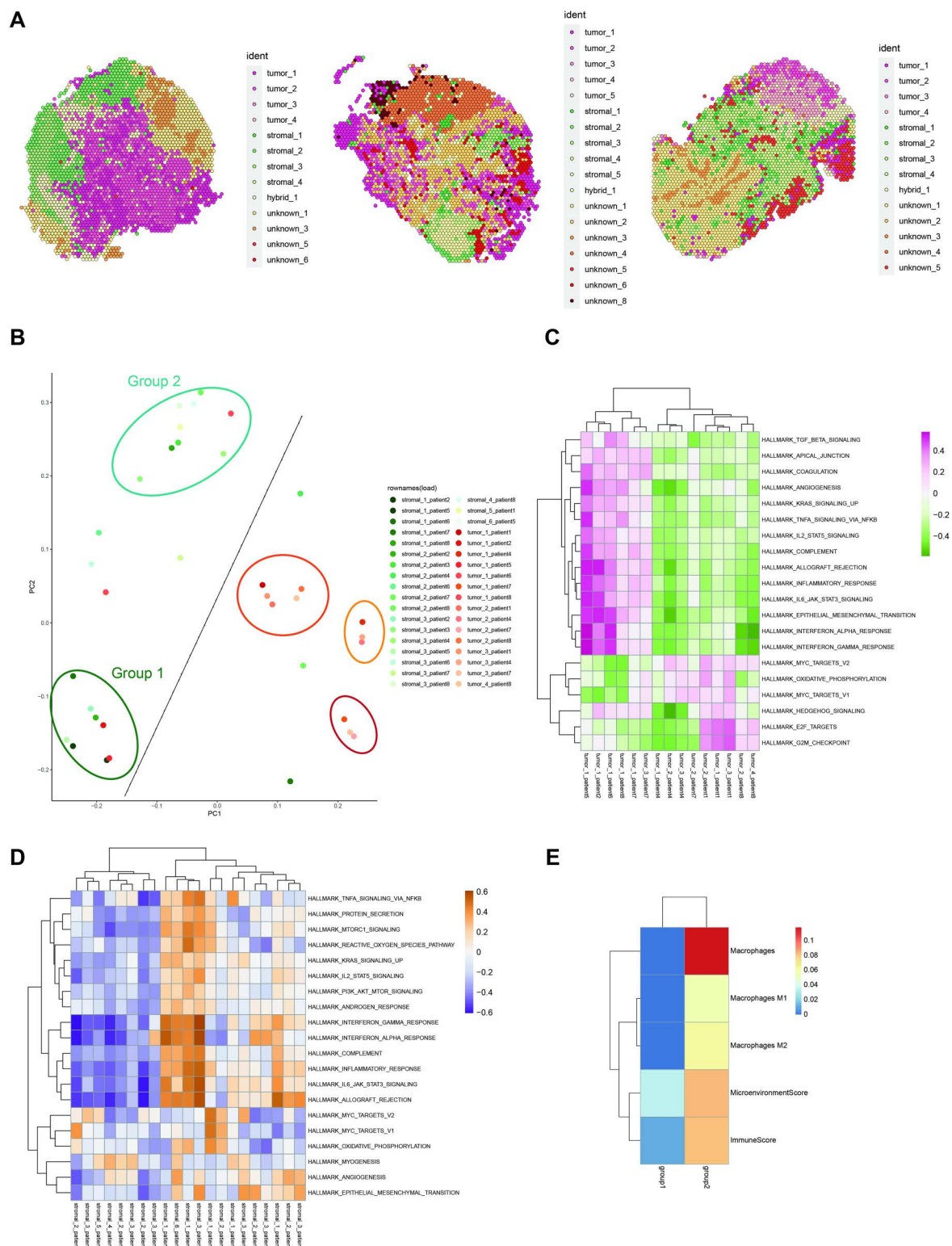

**Fig 2. Gene expression and pathway enrichment of tumor and stromal clusters.** (A): Spatial distribution of clusters in patients (From left to right: patient 1, patient 4, patient 8). (B): Visualization of clusters' gene expression through dimensional reduction. (C): Heatmap of pathway activity in tumor clusters. (D): Heatmap of pathway activity in stromal clusters. (E): Heatmap of cell enrichment in two groups (divided by the different enrichment of Gene Set Variation Analysis (GSVA) signaling pathways) of stromal clusters.

immunity. Additionally, when evaluating pathway activity by GO analysis through differentially expressed genes (DEGs), Group 2 showed strong enrichment in MHC class II-mediated immune responses, which are associated with T cell activation, indicating the presence of active T cell immunity in this stromal region (Fig S2B). In contrast, Group 1 lacked immune cell infiltration but was enriched in processes related to extracellular matrix development (Fig S2B), consistent with the GSVA results. This in-depth analysis illuminates the nuanced differences between these stromal groups, providing insights into their distinct roles and contributions to the complex landscape of HGSOC.

Additionally, our analysis revealed a separation of tumor clusters into three distinct groups. One group predominantly comprised clusters separate from those of patient_1 and patient_8, another group originated from patient_4, and the third emerged from patient_7. To unravel the intertumor heterogeneity within the tumor regions, we employed GSVA to calculate the activities of hallmark pathways based on the averaged expression of the selected tumor clusters. Hierarchical clustering of pathway activities across each tumor cluster revealed differences primarily dependent on significant enrichment in two groups of signaling pathways. The first group was associated with signaling pathways related to growth factor-regulated tumor growth and inflammatory response (group 1), and the second was linked to cellular proliferation in the G1 phase (group 2) (Fig 2C). Despite tumor clusters generally exhibiting higher enrichment in processes related to cellular proliferation and lower enrichment related to growth factor-related processes compared to the stromal group, there appeared to be a heterogeneous distribution within these two groups of pathways among all tumor cells.

To investigate the potential contributors to this molecular similarity, we explored CNVs in these clusters. Surprisingly, we did not identify common CNV patterns shared across entire groups, although some clusters within a group exhibited similarity. In group 1, two distinct CNV patterns emerged, one featuring significant addition in chromosomes 1, 6, 15, and 22, while the other involved losses in chromosomes 15 and 22 (Fig S2C). In contrast, group 2 contained 4 patterns, with clusters from the same sample displaying high similarity (Fig S2D). In summary, the CNV patterns of clusters with similar signaling pathway enrichment can vary. Further investigation is warranted to elucidate the underlying causes of this phenomenon.

## Investigating intratumor heterogeneity through inferCNV

As clonal evolution proceeds, the tumor accrues a diverse array of genetic alterations. This genetic diversity underlies the heterogeneity observed in High-Grade Serous Ovarian Carcinoma (HGSOC), encompassing variations in pathway activity and responses to treatment. Conducting CNV analysis enables the construction of a clonal evolution tree, facilitating the identification of distinct clones and shedding light on the process of clonal evolution within tumors.

To comprehend the intricacies of clone heterogeneity and how clonal evolution contributes to intratumor diversity, we selected two samples exhibiting varying degrees of malignancy. After conducting separate inferCNV analyses, we constructed clonal evolution trees that infer the progression of CNVs. Notably, we observed that tumors primarily originate from multiple CNVs rather than a single source (Fig 3A, B, C). Under the assumption that the earliest CNVs occurring in cells may be inherited by all their progeny, tumor cells originating from the same cellular source should share identical CNVs. However, in the two analyzed samples, we found that all tumor cells could be categorized into two distinct groups, suggesting that none of the tumor cells in these samples shared a common CNV origin. Similarly, in inpatient eight, tumor cells may have CNVs occurring across multiple chromosomes rather than a single one. For patient eight, one of its origins displayed additions in chromosomes 2, 3, 7, 8, 12, 13, 14, 20, and 21, along with losses in chromosomes 11, 17, and 19, indicating that multiple CNVs

may co-occur during tumor initiation (Fig 3C). The clonal evolution tree not only elucidates clone evolution but also categorizes spots into different groups based on their CNVs. To visualize the spatial relationship between distinct clones, we spatially mapped them. Tumor cells with similar origins may not necessarily be located nearby. For instance, in patient 4, clones G, P, and J shared the same origin but were dispersed across different regions.

To assess the impact of CNVs on tumor development, we evaluated the enrichment of pathway activity across different clones and conducted hierarchical clustering. Interestingly, the similarity in pathway activity was more closely associated with the spatial location of the clones rather than their CNVs, indicating that neighboring regions tended to exhibit higher similarity. Using the example of a patient eight sample, we observed that eight clones could be grouped into two modules, with one module related to cellular proliferation and metabolism, while the other was associated with epithelial-mesenchymal transition (EMT) and inflammatory response. This similarity in pathway activity appeared to be primarily influenced by the spatial location of the clones (Fig 3G). However, when we combined the results of GSVA with genes suspected of abnormal expression based on inferCNV, we were able to identify certain CNVs that might affect pathway activity (S3 Table). For instance, clones H, K, and M shared a similar loss in chromosome 17, leading to lower expression of a series of genes such as IGFBP4, GRB7, EZH1, ERBB2, EIF1, CAVIN1, ATP6V0A1, and AOC3, all located in the same region of chromosome 17 (chr17-region_143, chr17-region_201, chr17-region_242). As a result, pathways related to hypoxia, EMT, and estrogen response exhibited significantly lower enrichment. Similarly, although clones J and L, which shared similar CNVs, were classified into two distinct groups based on pathway activity, L still displayed some degree of similarity in the pathways highly enriched in J. Specifically, J and L shared similar additions in chromosome 11, involving ITFM and IRF7, both located in the same region of this chromosome (chr11-region_44, chr11-region_263). The pathways related to these genes, such as interferon response, complement, and apoptosis, were highly enriched in J and K. These findings suggest that specific genes inferred from CNVs may indeed impact pathway activity and contribute to tumor progression. Characterizing these genes could help uncover the drivers of tumor progression (Fig 3G, S3A, S3B)

We further investigate the impact of CNV heterogeneity on immune infiltration by performing MIA analysis in each clone. For these two patients, there is no significant enrichment of B cells across all samples, consistent with our MIA results. In patient 4, only clone G shows significant enrichment in macrophages, while other clones do not exhibit immune infiltration. According to the clonal tree (Fig 3A), clone G is unique and does not share any copy number variations (CNVs) with the other clones, which may explain its distinctive pattern of immune infiltration (Fig S8B). In contrast, patient 8 presents a different pattern. Clones derived from various branches, including H, J, L, and M, exhibit relatively high infiltration of both macrophages and T cells. However, clones generated from the same branch do not necessarily share similar immune infiltration profiles. For example, clones K and H come from the same branch as clones L and J, and clones I and N also originate from the same branch. This lack of correlation between immune infiltration and specific CNV events highlights the complexity of CNV and immune cell interactions in high-grade serous ovarian carcinoma (HGSOC).

## Illuminating Tumor Cell Heterogeneity through scRNA-Seq Analysis

To investigate the heterogeneity of the tumor cells, we initially employed inferCNV for the identification of tumor cells through scRNA-seq analysis (Fig S4G). To evaluate the heterogeneity of gene expression and function within the tumor, we defined the intratumor expression programs in the tumor cells of each sample using non-negative matrix factorization (NMF).

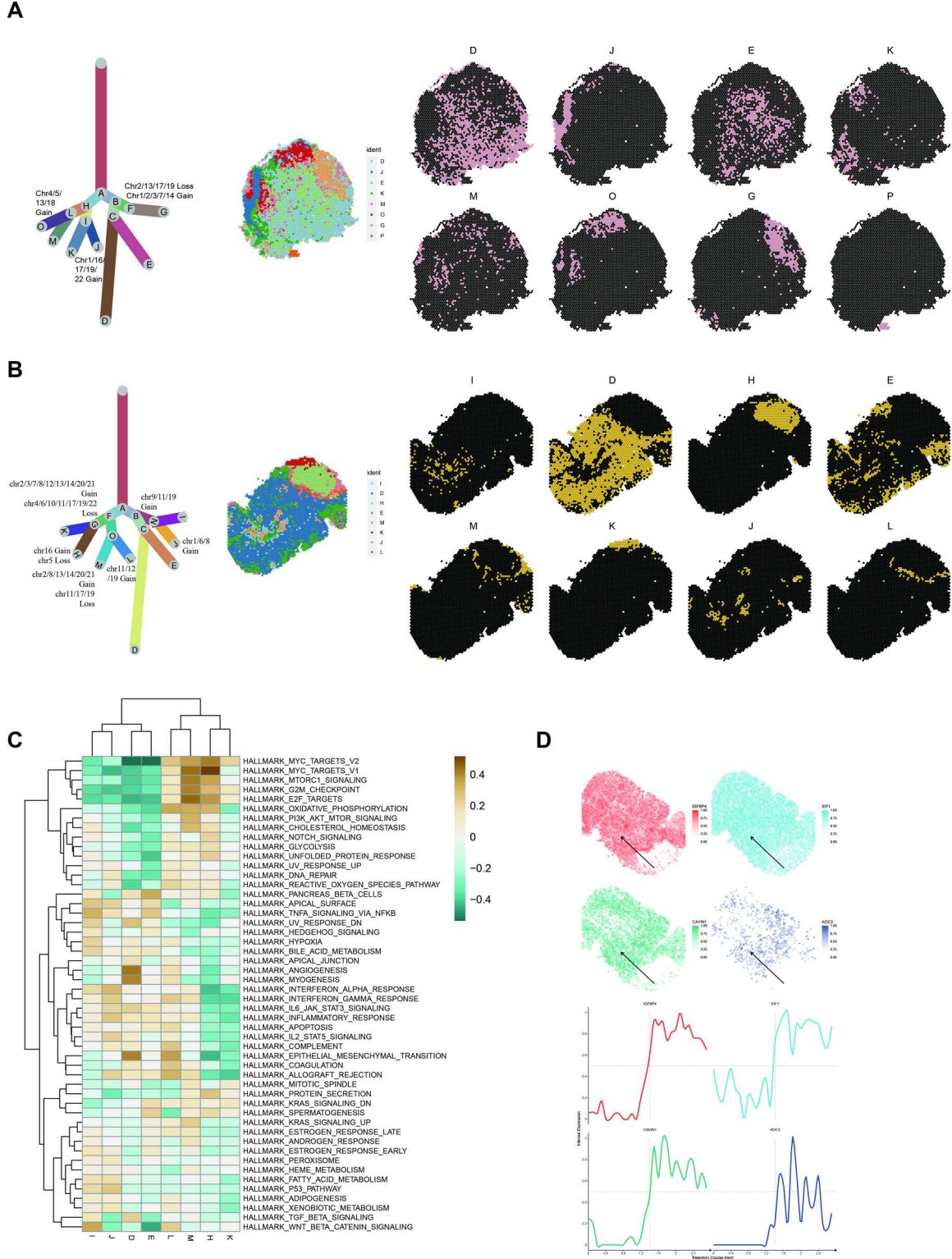

**Fig 3. Depict tumor heterogeneity based on copy number variation.** (A - B): Clonal evolutionary tree (left) and spatial visualization (middle and right) of patient 4 (A) and patient 8 (B). (C): Heatmap of pathway activity of different clones in patient 8. (D): Changes in gene expression potentially affected by copy number variations (CNVs) in patient 8.

In total, we identified 15 intratumor expression programs and determined three meta-programs shared by multiple tumors.

The modules, characterized by their top 50 scoring genes, had diverse functions, including epithelial differentiation (e.g., Keratin 1, Claudin-1), stress response (e.g., JUNB, FOS), major histocompatibility complex II (MHC-II) immune response (e. g., human leukocyte antigen - DR1 (HLA - DR1), HLA - DP1, CD74), and inflammation (e. g., IL6, IL33, tumor necrosis factor receptor superfamily member 12A (TNFRSF12A)) (Fig 4A). Meta-program 1 was distinguished by the expression of MHC-II-related genes such as CD74 and HLA - DRA and stress-response-related signature FOS and JUNB. Epithelial differentiation genes CLDN, KLK, and S100A1 were found in meta-program 2. Meta-program 3 comprised mitochondrial and ribosomal genes (Fig 4B).

The enrichment of these meta-programs in spatial distribution varied. The epithelial differentiation meta-program demonstrated widespread enrichment in most tumors, while MHC-II and stress-response meta-programs exhibited high enrichment in four of them. Meta-program 3 displayed significant enrichment exclusively in the tumor of patient_5, highlighting the heterogeneity in the composition of tumor cells within these tumors (Fig 6A). Spatial visualization further demonstrated the co-localization tendency of these meta-programs (Fig S3D).

Next, we integrated the scRNA seq data from different samples and performed sub-clustering, resulting in the identification of six clusters (Fig 4C). Two clusters exhibited high expression of the signatures of meta-program 1(Tumor 1 and 2), one showed elevated expression of the meta-program 2 signatures (Tumor 5), while another displayed high expression of meta-program 3 signatures (Tumor 0). Tumor 4 showed a hybrid expression of signatures from meta-program 2 and meta-program 3, and Tumor 1 remained unidentified (Fig 4D).

To unravel the interplay between cells and understand their roles in tumor progression, we conducted cell-cell communication analysis between clusters, revealing several ligand-receptor pairs. C-X-C motif chemokine ligand - atypical chemokine receptor 1 (CXCL - ACKR1) and CCL - ACKR1, chemokine signaling ligand-receptor pairs with positive prognosis values, were identified, suggesting a potential modulation of the immune response. The macrophage migration inhibitory factor (MIF), and its interacting partners CD74 and CD44 were highlighted, recognized for their roles in injury protection, healing promotion [38], and B cell survival [39], plays a pervasive role in mediating communications between tumor cells and macrophages. Ligand - receptor pairs midkine - syndecan 4 (MDK - SDC4) and MDK - nucleolin (NCL), which is associated with tumor growth and communication between tumor cells and cancer-associated fibroblasts (CAFs) in esophageal squamous cell carcinoma, were also observed[34]. The elevated expression of NCL has been associated with tumor progression and cell proliferation in breast cancer, both in vitro and in vivo [35]. While the precise effect of these ligand-receptor pairs in ovarian cancer remains unclear and requires further investigation, they represent potential drug targets indicated based on previous experiments (Figs 4E and 4F).

It's worth noticing that cell-cell communication also exhibited heterogeneity. For instance, Tumor 1, the cluster with MHC-II and stress-response signatures, displayed uniqueness in cell-cell communications. ACKR1 was exclusively expressed in the Tumor 1 cluster, suggesting a potentially pivotal role in regulating immune cell recruitment. Tumor clusters 2, 3, and 5 lacked receptors of tumor progression ligand-receptor MDK - SDC4 and exhibited lower expression of the receptors of MDK - NCL. CD44 is a drug target in Prostate Cancer [40] and Neuroblastoma [41], its prevalence in HGSOC tumor cells underscores its potential significance as a therapeutic target. In our HGSOC tumor sample, we found most tumor clusters lacked expression of CD44, except tumor cluster 4, the cluster with a hybrid high expression

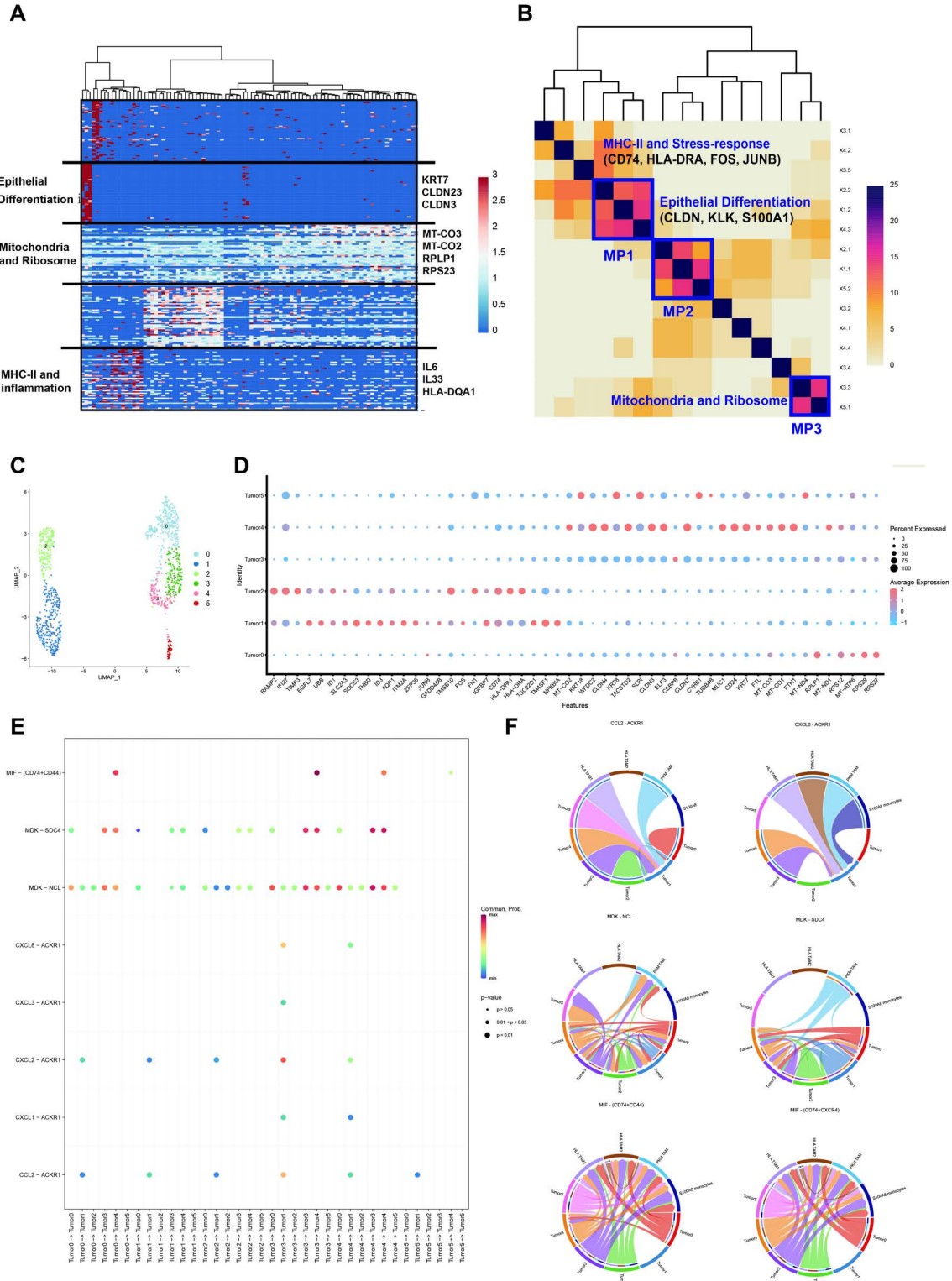

**Fig 4. Cellular meta-programs and cell-cell communication.** (A): Heatmap of expression programs in representative patients. (B): Shared meta-programs across different patients. (C): Sub-clustering of tumor cells. (D): Expression of meta-program signatures in tumor cell clusters. (E): Dot plot of cell-cell communication between tumor cells. (F): Chord plot of important ligand-receptor pairs in the communication of tumor cells.

of signatures from meta-program 2 and meta-program 3. Based on this, cluster 4 became the only group capable of receiving signaling sent by MIF, highlighting its potential role in regulating anti-tumor immune response.

## MDK-NCL Signaling Pathway Enhances Tumor Cell Proliferation

Our analysis also identified a highly enriched ligand-receptor pair MDK - NCL, contributing to communication among HGSOC tumor cells. NCL encodes a protein called nucleolin, primarily located in the nucleolus, and is involved in the regulation of transcription. Previous studies have reported its overexpression in multiple cancers, associating it with the promotion of tumor cell proliferation and tumor invasion[35–39]. Despite these findings, the specific function of NCL in HGSOC remained unclear. Given its prevalent role in HGSOC tumor cell communication and potential involvement in tumor malignancy, we hypothesize that the overexpression of NCL in HGSOC tumor cells may contribute to the promotion of tumor cell proliferation.

Previous experiments have suggested that the suppression of NCL could lead to the upregulation of phosphatase and tensin homolog (PTEN) and reduce the activation of AKT in breast cancer[35]. Hence, we posit that NCL overexpression may similarly promote tumor cell proliferation in HGSOC (Fig 5A). To test this hypothesis, we performed experiments using an HGSOC tumor cell line SKOV3. We generated a SKOV3 cell line with NCL overexpression (Fig 5C). Then we evaluated the effect of NCL overexpression on the AKT pathway using qPCR and Western Blot. Our findings revealed a significant upregulation of AKT in gene expression level, particularly at the protein level, the expression of pAKT is significantly higher in the NCL overexpression group, while AKT protein does not have a significant difference (Figs 5E, G, J, K, S4 Table). We also observed significant downregulation of PTEN in both gene expression and protein levels(Figs 5D, G, I, S9, S4 Table). These results indicate that NCL overexpression may promote the activation of the AKT pathway. The gene expression and protein level of Ki-67(Fig 5F, L, S4 Table) and the cell proliferation assay (Fig 5B) further confirmed this finding. We found that the expression of Ki-67 in both gene and protein and cell proliferation was significantly higher in the overexpression group compared to the control. In summary, our analysis identifies the high enrichment of MDK-NCL in HGSOC tumor cell communication, and subsequent experiments validate that the overexpression of NCL indeed promotes HGSOC tumor cell proliferation through the activation of the AKT pathway.

## Identification of the heterogeneity origin through inferCNV

Tumors typically develop as a result of various genetic mutations, with CNVs playing a significant role. Previous research has demonstrated that tumor cells often exhibit significant CNVs. Understanding that the diversity within tumors arises from the accumulation of mutations, investigating CNV heterogeneity can provide insights into the origin of tumor diversity. To delve into CNV heterogeneity, we initially characterized CNV patterns within each sample. This analysis revealed a spectrum of CNV patterns (Figs 6A, 6B, and S4 A - F).

In general, while samples exhibited certain similarities in gene expression analysis, their CNVs displayed substantial heterogeneity. The distribution of spots with CNVs varied, occurring in either numerous groups of spots or specific clusters, regardless of the malignancy degree. These findings underscore the heterogeneity of CNVs within the samples. To identify common CNVs shared among these patients and assess their heterogeneity, we integrated datasets from eight patients. Hierarchical clustering was applied to the CNV results, revealing a complex landscape(Fig 6A, S7). Surprisingly, no CNVs were identified that were shared by all eight patients, despite all samples being collected from HGSOC. This finding reveals

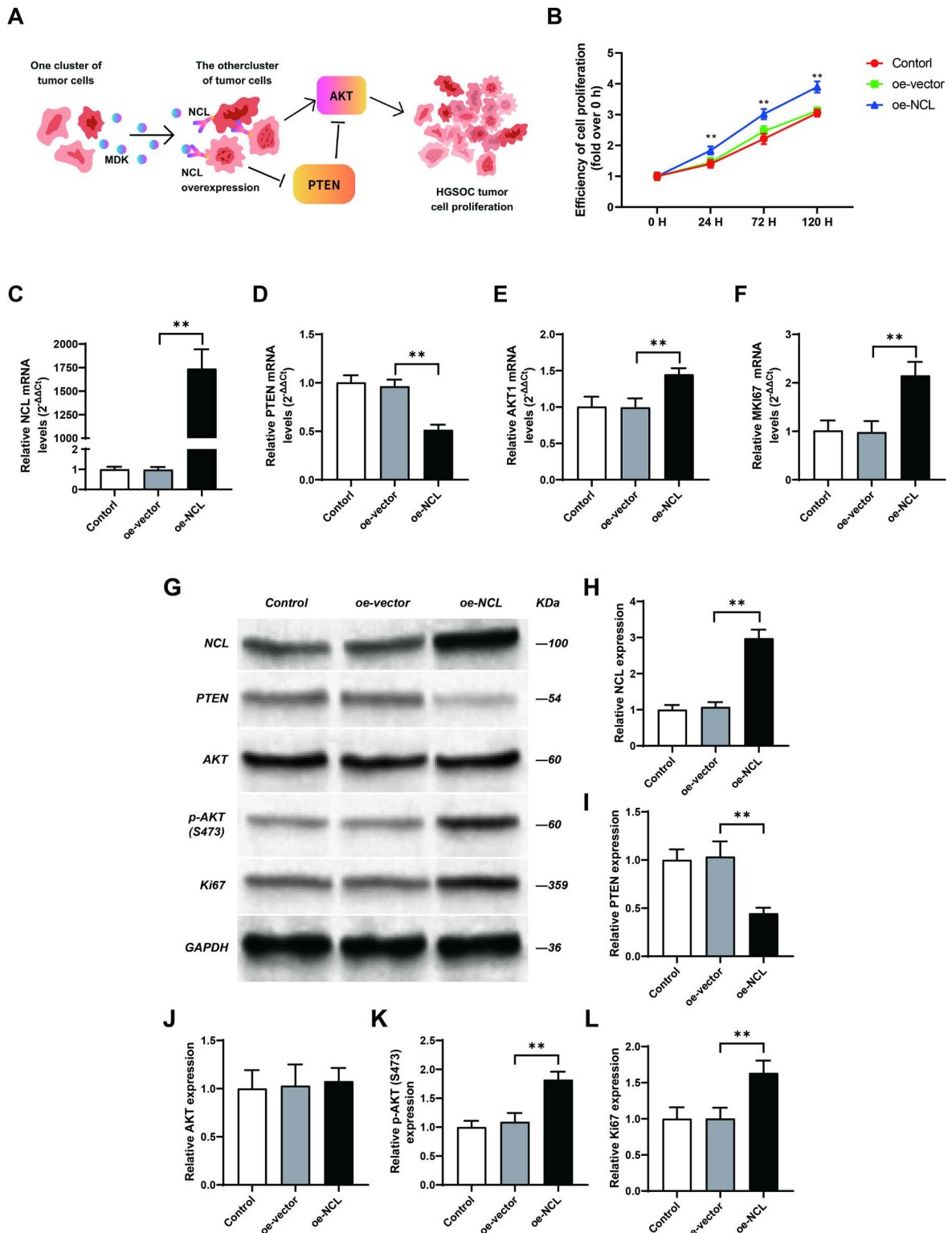

**Fig 5. NCL overexpression promotes tumor cells' proliferation.** (A): NCL overexpression activates the AKT pathway (B): Effect of NCL overexpression on SKOV3 cell proliferation (N = 3). SKOV3 cells were transfected with an NCL overexpression plasmid or the corresponding empty vector. Cell Counting Kit-8 (CCK8) assays were utilized to evaluate cells' proliferation in each group at different time points. Results are presented as mean ± standard deviation (SD) from three independent experiments. ** indicates P < 0.01. (C - F): Effect of NCL overexpression on gene expression(N = 3). The gene expressions of NCL (C), phosphatase and tensin homolog (PTEN)

(D), AKT1 (E), and Ki-67 (F) in each group of cells were examined using quantitative polymerase chain reaction (qPCR). Results represent the mean ± SD from three independent experiments. ** indicates P < 0.01. (G - L): The effect of NCL overexpression on protein level. The protein expression levels of NCL (G, H), PTEN (G, I), AKT (G, J), p-AKT (S473) (G, K), and Ki67 (G, L) in the cells from each group were detected by Western Blot. Results are presented as mean ± SD from three independent experiments. **P < 0.01.

the remarkable heterogeneity of HGSOC in terms of CNV levels, highlighting the molecular diversity within this cancer subtype and offering a potential explanation for its overall heterogeneity.

Moreover, the CNV pattern tends to be sample-specific, indicating that different samples have unique CNV profiles. Although no CNVs were shared among all eight samples, some were shared by subsets of tumors. For instance, the loss of chromosome 1 was shared by patient_4, patient_7, and patient_8; the loss in chromosome 8 was shared by patient_1, patient_4, patient_7, and patient_8; and the loss in chromosome 22 was shared by patient_1, patient_4, and patient_7. These results further highlighted the heterogeneous nature of CNV patterns in HGSOC. Further investigation into the relationship between CNVs and clinical outcomes is warranted. The CNV scores from each sample were visually represented, and the enrichment of various cell types was assessed using cell signatures (Figs 6B, S5, and S6). Tumor regions were defined by calculating the geometric mean of CNV scores and tumor signatures, providing a comprehensive view (Fig 6C).

## Identification of tumor-associated macrophages

Tumor-associated macrophages (TAMs) play a critical role in shaping the tumor microenvironment by actively participating in immune and inflammatory responses. Previous reports have highlighted a correlation between poor prognosis in ovarian cancer and an overrepresentation of anti-inflammatory TAMs. Building upon our earlier findings, wherein one stromal cluster exhibited a high enrichment of macrophages coupled with elevated immune and inflammatory features, we sought a deeper understanding of the specific types of macrophages and their interactions with tumor cells using scRNA seq.

To elucidate the diverse macrophage subpopulations, we conducted sub-clustering on all macrophages within the scRNA-seq dataset, revealing the existence of four distinct clusters (Fig 7A). Cluster 4 was identified as S100A8 monocytes, characterized by high expression of CD14 and S100A transcripts, aligning with previous research associating these markers with monocytes. The remaining clusters, exhibiting high expression of CD83, and HLA-DQ signatures, were designated as TAMs (Fig 7B). Clusters 0 and 1, distinguished by the differential expression of HLA-DQ, HLA-DR, and apolipoprotein E (APOE) genes, were denoted as HLA high immuno-suppressive subtypes, specifically labeled as HLA TAM1 and HLA TAM2, respectively. Cluster 2 was annotated as PKM TAM since PKM is one of the differential expressed genes in this cluster. To gain insights into the heterogeneity of macrophages and their interaction with tumor cells, we visualized the enrichment of different macrophage groups and tumor cells in different patients. The observed heterogeneity in macrophage enrichment suggests that the immune infiltration in the ovary varies between different patients (Fig 7C).

The analysis of cell-cell communication has revealed multiple signaling pathways involved in the dynamic interaction between macrophages and tumor cells (Figs 6E and 6F; S3C and 3E). Notably, MIF played a crucial role in tumor immune responses through interactions such as MIF - CD74/CD44. This interaction was observed in almost all cell-cell communications between tumor cells and macrophages. Disrupting this pathway could significantly impact the tumor microenvironment. However, it is important to highlight that the absence of these

**A**

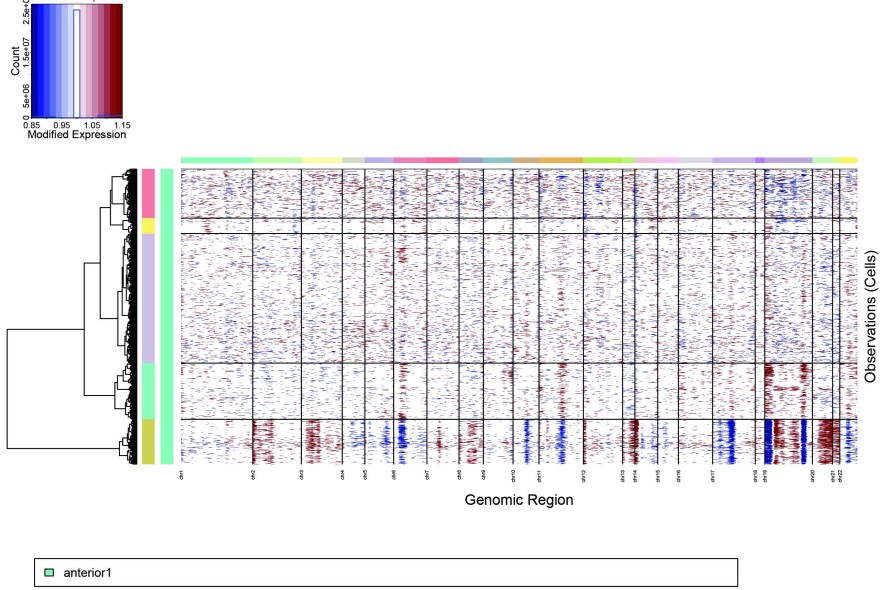

**B**

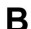
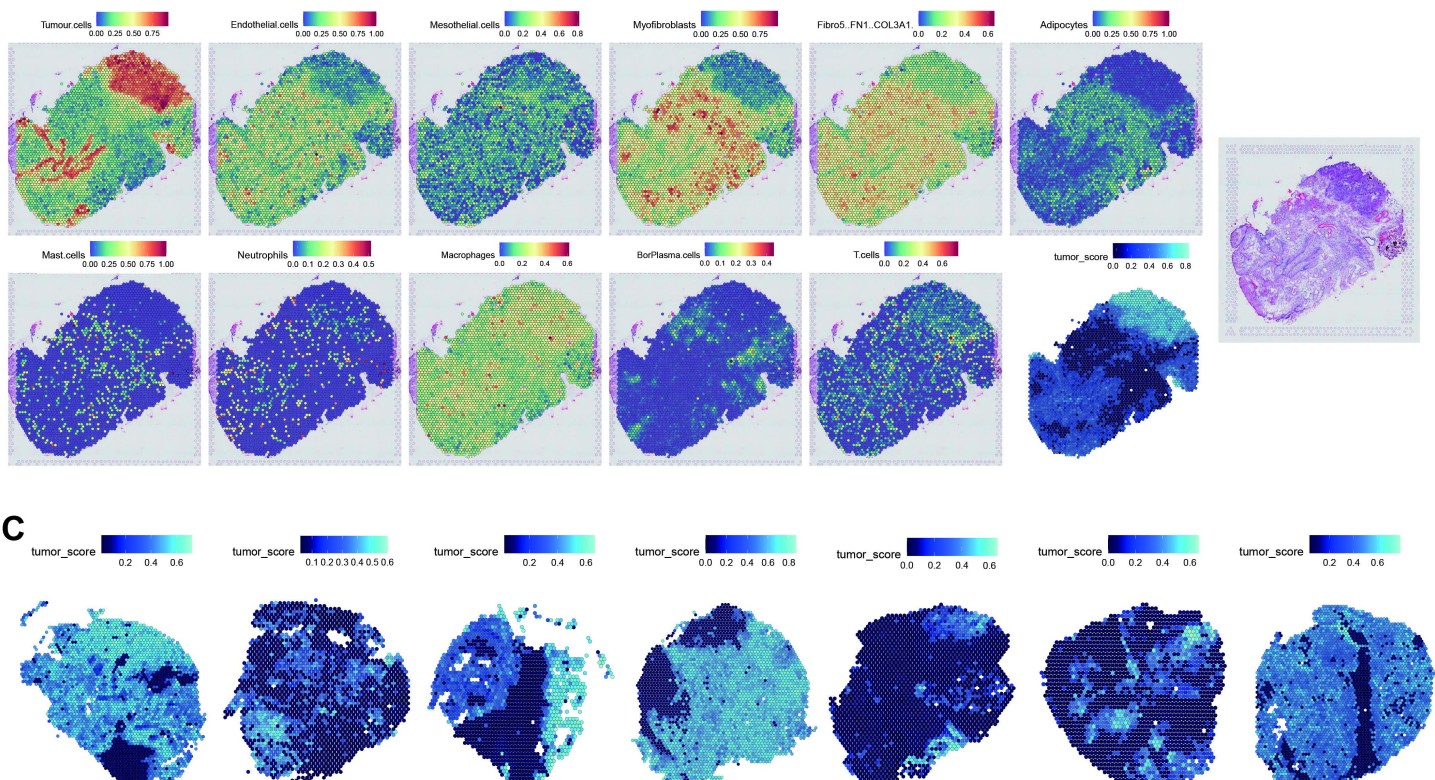

**Fig 6. Cell type enrichments in different patients.** (A): Heatmap of copy number variations (CNVs) in patient 8 (B-C): Enrichment of cell type(B) and tumor score(C). Normalized enrichment of different cell types and tumor score (blue and black) in patient 8 (left). Hematoxylin and Eosin (H&E) staining image (right).(D): Tumor score in patients 1–8.

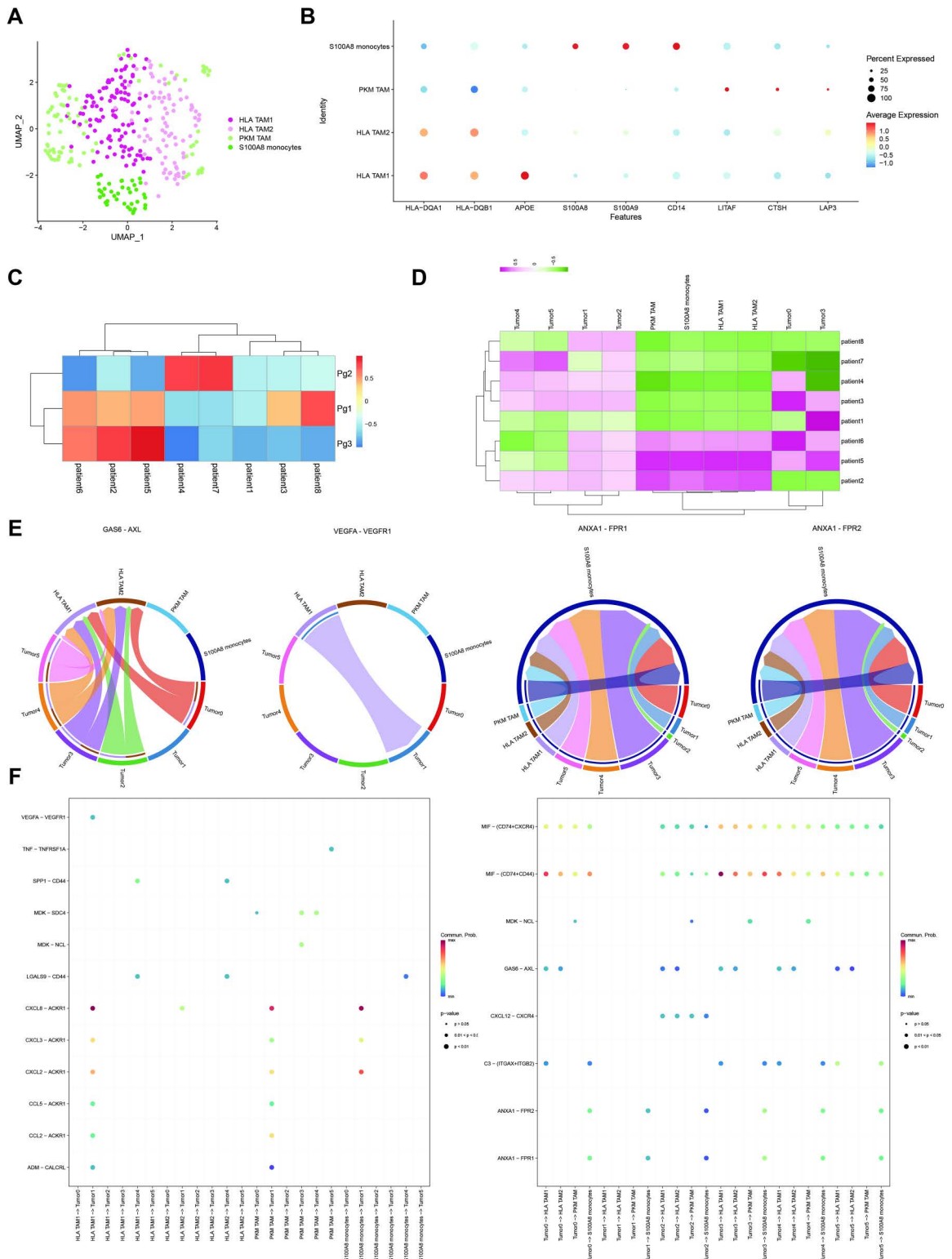

**Fig 7. Analysis of macrophages in patients.** (A): Sub-clustering of macrophages. (B): Dot plot illustrating the expression of signatures. (C): Enrichment of different meta-programs across patients. (D): Enrichment of different groups of tumor cells and macrophages across patients. (E): Chord plot depicting important ligand-receptor pairs in cellular communication. (F): Dot plot visualizing cell-cell communication between tumor cells and macrophages.

ligand-receptor pairs between tumor cluster 2 and macrophages may lead to distinct drug responses in this subset of tumor cells compared to others.

In addition to their involvement, ligand-receptor pairs also exhibited heterogeneity across different clusters, including those associated with positive prognosis (CXCL - ACKR1/CCL - ACKR1) and those linked to poorer outcomes (vascular endothelial growth factor A(VEGFA) - vascular endothelial growth factor receptor 1 (VEGFR1), growth arrest-specific 6 (GAS6) - AXL receptor tyrosine kinase (AXL), Annexin A1 (ANXA1) - formyl peptide receptor (FPR)). The CXCL - ACKR1/CCL - ACKR1 pairs potentially contributed to the enhanced immune response within the tumor. Notably, these pathways exhibited specificity in their interactions. For instance, while all macrophages could act as senders, only tumor cluster 1 had the receptors for CXCL - ACKR1/CCL - ACKR1 ligand-receptor pairs. The ACKR1 receptor, associated with chemokine regulation and anti-tumor responses [47], aligns with cluster 1's MHC II character, as indicated by the MPs' signatures. AXL in GAS6 - AXL ligand-receptor pairs, implicated in HGSOC drug resistance [48], was exclusively expressed in HLA TAM1 and HLA TAM2, and only monocytes could communicate with tumor cells through ANXA1–FPR, which serve as effective mediators in controlling inflammatory responses [49]. The VEGFA - VEGFR1 interaction only occurred between HLA TAM1 and tumor cluster 1. VEGFR-1 activation in TAMs has been associated with cancer immune evasion through immunosuppressive cytokine release [50]. These unique ligand-receptor pairs in different macrophage clusters strongly suggested that blocking specific ligand-receptor pairs could selectively disrupt communication between particular tumor cell groups and macrophages, potentially offering new drug targets and elucidating patient responses to therapies.

## Discussion

MDR poses an immense challenge in the treatment of HGSOC, and its association with cancer heterogeneity has long been recognized [1]. While previous research has made significant progress in delineating heterogeneity within tumor cells through scRNA-seq [11–15], the spatial heterogeneity of HGSOC has remained a relatively underexplored domain. This study addresses this gap by integrating ST and scRNA-seq data, offering a comprehensive analysis of HGSOC tumor heterogeneity. Our investigation of spatial heterogeneity encompasses the characterization of tumor clusters, cellular composition, and signaling pathways, all grounded in gene expression profiles.

Tumor heterogeneity, a prominent feature of cancer, is intricately intertwined with CNVs, and decoding these CNV patterns is crucial for a deeper understanding of heterogeneity development. In our study, we leveraged the inferCNV tool to delineate CNVs within ST data. Although this tool has primarily been used to identify malignant cells [18,40–42], our application extended its utility to innovative domains, including constructing clonal evolution trees [17] and evaluating the heterogeneity of tumor cells [19]. Our investigation sought to determine both homogeneous and heterogeneous CNV patterns across HGSOC, with the ultimate goal of deciphering the origins of heterogeneity. Additionally, we aimed to provide insights into the underlying causes of heterogeneity in signaling pathways. Crucially, our study delved into the spatial heterogeneity of HGSOC through the lens of CNV clones. Each clone represented a distinctive branch in the evolutionary trajectory of CNVs during tumor development. By juxtaposing gene expression, signaling pathways, and spatial locations of these clones, we dissected the intricacies of their heterogeneity. We also unearthed associations between abnormal gene expression influenced by CNVs and pathway enrichment, thereby offering plausible explanations for specific patterns of pathway activity.

In summary, our study pioneered the exploration of tumor heterogeneity through the lens of CNV clones within ST data, a dimension that has received limited attention in prior

research, particularly when employing ST data. Our work not only elucidates the spatial distribution and correlations among CNV clones but also advances our understanding of clonal evolution within a spatial context. Furthermore, our approach to investigating heterogeneity through CNV clones offers a biologically relevant avenue for tumor heterogeneity research.

scRNA-seq has emerged as an invaluable tool, enabling a comprehensive dissectiion of the intricate landscape of cellular diversity within tumor microenvironments. Our investigation unveils three fundamental cellular programs shared across tumor cells, encompassing immune response, epithelial differentiation, and metabolic processes. Subsequent sub-clustering efforts unveil five distinct tumor cell clusters and three macrophage clusters, providing a nuanced understanding of the heterogeneity within the tumor microenvironment.

Through cell-cell communication analysis, we identified highly enriched ligand-receptor pairs facilitating communication between different clusters of tumor cells. For example, MDK - NCL as the most prevalent ligand-receptor pair mediating communications among tumor cells. This ligand-receptor pair, previously identified in scRNA-seq analyses of HGSOC tumor cells, is associated with communication between tumor cells and CAFs, a phenomenon previously reported in ovarian cancer and esophageal squamous cell carcinoma [34]. In endometrial carcinoma, its heightened expression across epithelial cells suggests a potential contribution to an immunosuppressive tumor microenvironment [39].

NCL has been known to regulate polymerase I transcription and has been reported overexpressed in multiple cancers. This overexpression has been associated with the activation of oncogenes, including interleukin-9 receptor (IL-9R) transcription in leukemogenesis [36], promotion of angiogenesis via VEGF [37], and E6 and E7 oncogene expression in human papillomavirus 18 (HPV18) [38]. Notably, the upregulation of NCL has been associated with the activation of phosphoinositide 3-kinase (PI3K) and protein kinase C zeta (PKCz) kinases, contributing to increased cell proliferation [43]. Previous experiments have reported that suppression of NCL may lead to the upregulation of PTEN and reduce the activation of AKT in breast cancer [35]. However, NCL's role varies across different cancers, and the mechanisms underlying its overexpression-induced HGSOC cell proliferation remain unclear. Several drugs targeting NCL have been developed, such as AS1411[44,45], which targets nucleoplasmic nucleolin, and HB-19[46,47], which targets cell surface nucleolin. These molecules have shown efficacy in suppressing tumor progression in breast cancer, positioning NCL as a potential therapeutic target for high-grade serous ovarian cancer (HGSOC).

Therefore, we propose that NCL overexpression may promote tumor cell proliferation via the AKT signaling pathway, and our experiments provide strong support for this hypothesis. Our findings show that inhibiting NCL expression reduces the rate of cell proliferation, offering new insights into potential clinical therapies. Our study did not directly examine the role of the MDK-NCL interaction in tumor proliferation but only focused specifically on NCL, which has been infrequently studied in this context. Previous research suggests that MDK activates survival pathways (via AKT, mTOR, and Bad) and inhibits senescence or apoptosis (by suppressing caspase-3), thereby contributing to chemoresistance [48]. Nucleolin, encoded by the NCL gene, serves as the receptor for MDK and plays a crucial role in mediating this process [49]. However, research into the downstream mechanisms of NCL remains limited. We propose that NCL may function through the classical PIK3-AKT signaling pathway [50], potentially mediated by the established regulatory mechanisms by previous research about MDK (Fig 8) [48]. While there is some research on the upstream regulation of the PIK3-AKT pathway, the findings are inconclusive regarding its role in upregulating AKT, and thus we have not included those studies in this discussion [51].

Identifying drug targets within molecules involved in cell-cell communication is an increasingly popular strategy. For instance, GATA Binding Protein 3 (GATA3) has been

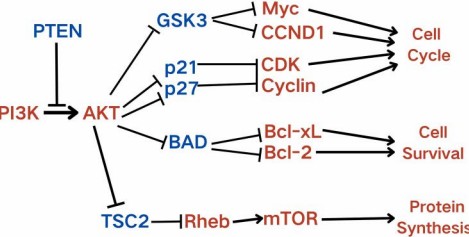

**Fig 8. Potential signaling pathways regulated by enriched MDK-NCL.** The potential pathways contribute to tumor progression regulated by the AKT pathway when MDK-NCL is highly enriched. Red represents up-regulate/activated, and blue represents down-regulated.

shown to contribute to tumor progression by promoting stemness, enhancing tumor development and metastasis, and inducing macrophage polarization in HGSOC, largely due to its release by tumor-associated macrophages and is a potential drug target for HGSOC therapy[52]. Similarly, MDK-NCL also has the potential to influence tumor progression; however, its role and mechanisms have been underexplored in this context. More research is needed to understand NCL's involvement in tumor pathogenesis. Additionally, GATA3 has been linked to tumor microenvironment (TME) dynamics, particularly its association with increased cancer-associated fibroblast (CAF) infiltration, which plays a crucial role in epithelial-mesenchymal transition and metabolically supports cancer cell proliferation[53]. Our study has primarily focused on the role of MDK-NCL in ovarian cancer cells, but further investigation into the role of MDK-NCL within the TME is warranted.

Our study provides a unique perspective on the spatial heterogeneity of HGSOC and introduces an innovative approach for analyzing this heterogeneity through CNVs. However, it is essential to acknowledge several limitations in our research that warrant consideration in future investigations. Firstly, our scRNA-seq analysis was conducted on a subset of samples, potentially resulting in the omission of unique cell populations present in the sequenced samples. Secondly, our description of the association between CNVs and heterogeneity remains qualitative, necessitating a quantitative analysis to establish a more robust correlation. Lastly, the connection between the molecular heterogeneity and clinical outcomes we have uncovered requires further exploration. In summary, our study provides a comprehensive exploration of HGSOC heterogeneity, including intertumor, intratumor, and cellular aspects through gene expression, CNVs, and single-cell analyses. This work contributes significantly to our comprehension of HGSOC heterogeneity, offering novel insights into the ovarian cancer landscape and holding promise for the development of personalized treatment strategies.

## Supporting information

**S1 Fig. Fig S1-S9.**
(PDF)

**S1 Table. Table S1the markers of each cluster in ST data.**
(CSV)

**S2 Table. Table S2 the markers of cell types calculated by scRNA seq data.**
(CSV)

**S3 Table. Table S3 the functional genes that may be affected by CNVs in patient 8.**
(CSV)

**S4 Table. Table S4 Experiment result of qPCR, CCK8 and Western Blot.**
(XLSX)

## Author contributions

**Conceptualization:** Songyun Li, Zhuo Wang.

**Data curation:** Songyun Li.

**Formal analysis:** Songyun Li.

**Funding acquisition:** Zhuo Wang, Hsien-Da Huang.

**Investigation:** Songyun Li, Zhuo Wang.

**Methodology:** Songyun Li, Zhuo Wang.

**Project administration:** Zhuo Wang, Hsien-Da Huang.

**Resources:** Zhuo Wang.

**Supervision:** Zhuo Wang, Hsien-Da Huang.

**Validation:** Songyun Li.

**Visualization:** Songyun Li.

**Writing – original draft:** Songyun Li.

**Writing – review & editing:** Zhuo Wang, Hsien-Da Huang.

## Acknowledgments

The authors wish to express their gratitude to TopEdit (www.topeditsci.com) for their valuable linguistic assistance during the process of preparing this manuscript and BIORN for the wet lab experiment conduction.

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
