## [Decision Letter · Decision Letter 0]

27 Sep 2024

PONE-D-24-32439Deciphering Ovarian Cancer Heterogeneity through Spatial Transcriptomics, Single-cell Profiling, and Copy Number VariationsPLOS ONE

Dear Dr. wang,

Thank you for submitting your manuscript to PLOS ONE. After careful consideration, we feel that it has merit but does not fully meet PLOS ONE’s publication criteria as it currently stands. Therefore, we invite you to submit a revised version of the manuscript that addresses the points raised during the review process.

We look forward to receiving your revised manuscript.

Kind regards,

Amr Ahmed El-Arabey

Academic Editor

PLOS ONE

Journal Requirements:

3. For studies reporting research involving human participants, PLOS ONE requires authors to confirm that this specific study was reviewed and approved by an institutional review board (ethics committee) before the study began. Please provide the specific name of the ethics committee/IRB that approved your study, or explain why you did not seek approval in this case.

5. Thank you for stating in your Funding Statement: 

"This work is supported by the Warshel Institute for Computational Biology funding 

from Shenzhen City and Longgang District; Natural Science Foundation of Guangdong 

(2023A1515011861); National Natural Science Foundation of China (No. 32070674); 

Shenzhen Science and Technology Program (JCYJ20220530143615035)."

Reviewers' comments:

Reviewer's Responses to Questions

**Comments to the Author**

1. Is the manuscript technically sound, and do the data support the conclusions?

Reviewer #1: Yes

Reviewer #2: Yes

2. Has the statistical analysis been performed appropriately and rigorously? 

Reviewer #1: Yes

Reviewer #2: Yes

3. Have the authors made all data underlying the findings in their manuscript fully available?

Reviewer #1: Yes

Reviewer #2: Yes

4. Is the manuscript presented in an intelligible fashion and written in standard English?

Reviewer #1: Yes

Reviewer #2: Yes

5. Review Comments to the Author

Reviewer #1: the manuscript is interesting, generally well written and illustrated. However, some points deserve to be improved. In particular:

Introduction, lines 1-7: it deserves to be pointed out that chemotherapy resistance in ovarian cancer cells is mainly due to the increased antioxidant capacity of these cells. This mechanism is particularly efficient against platinum-derived chemotherapeutics (see PMID: 38203758, PMID: 37175546).

Introduction, Lines 4-10: referenes are needed

Plasmid and Cell Transfection: transfection time and the amount of plasmid used must be reported

Figure 6: images in A,B and C are unreadable

Authors must add the number of replicates (N) in the figures' legends

Figure 5: in oder to state that "NCL overexpression activates the ATK pathway", authors must test the phospho-AKT expression by western blot analysis since the only increase of AKT mRNA does not mean that is also phosphorylated (the active form of AKT).

Abbreviations must be written in full length when mentioned for the first time

Reviewer #2: The purpose of this work was to provide a comprehensive perspective of tumor heterogeneity across the range of gene expression, copy number variation (CNV), and single-cell profiles. The current proposal is interesting and well-written. Therefore, I recommend that the current study be published after major revisions as follows:

1- Could the authors provide the impact of heterogeneity across the spectrum of gene expression, copy number variation (CNV) on the infiltration of immune cells for HGSOC patients?

2- Please discuss the following articles:

Multitarget strategy of GATA3 and high-grade serous ovarian carcinoma: Where are we now? Thromb Res. 2024 Apr;236:1-3. doi: 10.1016/j.thromres.2024.02.013.

GATA3 as a regulator for naughty cancer-associated fibroblasts in the microenvironment of high-grade serous ovarian cancer. Hum Cell. 2021 Nov;34(6):1934-1936. doi: 10.1007/s13577-021-00598-w.

3- Please add a diagrammatic figure to propose the possible mechanistic pathway for these findings

4- Please check the manuscript for typos and error

6. PLOS authors have the option to publish the peer review history of their article (what does this mean? ). If published, this will include your full peer review and any attached files.

**Do you want your identity to be public for this peer review?** For information about this choice, including consent withdrawal, please see our Privacy Policy .

Reviewer #1: No

Reviewer #2: **Yes: ** Mohnad abdalla

---

## [Author Response · Author response to Decision Letter 1]

27 Nov 2024

Note: All the content of the response letter here contained in the cover letter, the file's name is called Response to Academic Editor and Reviewers. The response letter contains some figure that cannot be shown here, so please check the uploaded response letter file

Response to Academic Editor and Reviewers

Manuscript Number: PONE-D-24-32439

Manuscript Title: Deciphering Ovarian Cancer Heterogeneity through Spatial Transcriptomics, Single-cell Profiling, and Copy Number Variations

Authors: Songyun Li, Zhuo Wang*, Hsien-Da Huang*

Dear Academic Editor and Reviewer(s),

Thank you very much for the letter containing valuable comments on our manuscript entitled “Deciphering Ovarian Cancer Heterogeneity through Spatial Transcriptomics, Single-cell Profiling, and Copy Number Variations”. These comments are very important for our study's improvement and further development. We want to express our sincerest gratitude to you and the reviewers who have spent a great deal of time and effort in evaluating our manuscript. We have revised the manuscript according to the reviewers’ suggestions, taking great consideration of the reviewers’ comments. I am submitting the revised manuscript along with a response letter indicating the changes we have made. All of the changes in the revised manuscript are marked in blue, accordingly.

We look forward to receiving a favorable reply.

Thank you！

Best regards,

Hsien-Da Huang,

Presidential Chair Professor,

Executive Deputy Director, Warshel Institute of Computational Biology,

Associate Dean, School of Medicine,

The Chinese University of Hong Kong, Shenzhen

Tel:+86 0755 23519601.

E-mail: huanghsienda@cuhk.edu.cn

Reviewers’ Comments and the Authors’ Revisions

Reviewer #1:

the manuscript is interesting, generally well-written, and illustrated. However, some points deserve to be improved. In particular:

1. Introduction, lines 1-7: it deserves to be pointed out that chemotherapy resistance in ovarian cancer cells is mainly due to the increased antioxidant capacity of these cells. This mechanism is particularly efficient against platinum-derived chemotherapeutics (see PMID: 38203758, PMID: 37175546).

Response: Thank you so much for your valuable advice, that’s very impressive! The revised version is shown as follows:

High-grade serous ovarian carcinoma(HGSOC), characterized by its aggressive behavior and rapid proliferation, is the most prevalent and lethal form of ovarian cancer, accounting for over 70% of ovarian cancer-related fatalities[1]. One of the primary challenges in treating high-grade serous ovarian cancer (HGSOC) is multidrug resistance (MDR), where cancer cells develop resistance to various chemotherapy agents, particularly platinum-based and taxane therapies, resulting in treatment failure[2]. Platinum-derived drugs generate reactive oxygen species (ROS), leading to DNA damage and apoptosis. However, HGSOC cells bolster their antioxidant defenses by upregulating antioxidants such as glutathione and NAD(P)H: quinone oxidoreductase 1, which contributes to their chemoresistance[3], [4].

2. Introduction, Lines 4-10: references are needed

Response: Thanks for your suggestions. The reference has been added shown as follows:

High-grade serous ovarian carcinoma(HGSOC), characterized by its aggressive behavior and rapid proliferation, is the most prevalent and lethal form of ovarian cancer, accounting for over 70% of ovarian cancer-related fatalities[1]. One of the primary challenges in treating high-grade serous ovarian cancer (HGSOC) is multidrug resistance (MDR), where cancer cells develop resistance to various chemotherapy agents, particularly platinum-based and taxane therapies, resulting in treatment failure[2]. Platinum-derived drugs generate reactive oxygen species (ROS), leading to DNA damage and apoptosis. However, HGSOC cells bolster their antioxidant defenses by upregulating antioxidants such as glutathione and NAD(P)H: quinone oxidoreductase 1, which contributes to their chemoresistance[3], [4]. MDR is caused by the inconsistent response of tumor cells to treatment, a variability rooted in the diverse and heterogeneous nature of tumor cells[5], [6]. Tumor heterogeneity refers to the diversity and variability of cells within a single tumor or among different tumors of the same type[7]. This diversity plays a crucial role in tumor progression and influences the response to treatment.

[1] Roberts, Cardenas, and Tedja, “The Role of Intra-Tumoral Heterogeneity and Its Clinical Relevance in Epithelial Ovarian Cancer Recurrence and Metastasis,” Cancers, vol. 11, no. 8, p. 1083, Jul. 2019, doi: 10.3390/cancers11081083.

[2] L. Wang et al., “Drug resistance in ovarian cancer: from mechanism to clinical trial,” Mol. Cancer, vol. 23, no. 1, p. 66, Mar. 2024, doi: 10.1186/s12943-024-01967-3.

[3] S. Fantone et al., “Role of SLC7A11/xCT in Ovarian Cancer,” Int. J. Mol. Sci., vol. 25, no. 1, p. 587, Jan. 2024, doi: 10.3390/ijms25010587.

[4] G. Tossetta, S. Fantone, G. Goteri, S. R. Giannubilo, A. Ciavattini, and D. Marzioni, “The Role of NQO1 in Ovarian Cancer,” Int. J. Mol. Sci., vol. 24, no. 9, p. 7839, Apr. 2023, doi: 10.3390/ijms24097839.

[5] I. Dagogo-Jack and A. T. Shaw, “Tumour heterogeneity and resistance to cancer therapies,” Nat. Rev. Clin. Oncol., vol. 15, no. 2, pp. 81–94, Feb. 2018, doi: 10.1038/nrclinonc.2017.166.

[6] A. Zhang, K. Miao, H. Sun, and C.-X. Deng, “Tumor heterogeneity reshapes the tumor microenvironment to influence drug resistance,” Int. J. Biol. Sci., vol. 18, no. 7, pp. 3019–3033, 2022, doi: 10.7150/ijbs.72534.

[7] D. Pe’er, S. Ogawa, O. Elhanani, L. Keren, T. G. Oliver, and D. Wedge, “Tumor heterogeneity,” Cancer Cell, vol. 39, no. 8, pp. 1015–1017, Aug. 2021, doi: 10.1016/j.ccell.2021.07.009.

3. Plasmid and Cell Transfection: transfection time and the amount of plasmid used must be reported

Response: Thanks for your suggestion, here is the revised version, which is added to the method section:

Plasmid and Cell Transfection

The full-length NCL gene was cloned into a pcDNA 3.1 (+) vector, incorporating ACC65I and EcoRI restriction sites. The primer sequences used for cloning NCL were: 5’-CGGGTACCATGGTGAAGCTCGCGAAGGC-3’ and 5’-GGGAATTCCTATTCAAACTTCGTCTTCTTTCCTTGT-3’.

For transfection, cells were plated in a 24-well plate at approximately 70% confluency the day prior to the procedure and maintained under standard culture conditions. Transfections were carried out using 500 ng of plasmid DNA and 1.5 μL of Lipofectamine 3000 mixed with 150 μL of OPTI-MEM medium, incubated at room temperature for 5 minutes. This mixture was then combined with an additional 150 μL of OPTI-MEM containing the plasmid DNA and incubated for an additional 15 minutes in a sterile environment. Prior to transfection, the culture medium was removed from each well, and cells were rinsed three times with 1 mL of 1× PBS. Next, 400 μL of OPTI-MEM was added to each well, and the plates were returned to a 37°C incubator with 5% CO2. After 15 minutes, the transfection mixture (RNA and plasmid) was added according to experimental groupings, gently mixed, and incubated at 37°C for 6 hours. Following this incubation, the medium was replaced with fresh serum-free medium. Transfection efficiency was assessed using PCR.

4. Figure 6: images in A, B, and C are unreadable

Response: Thanks for your advice. Figure 6 has been modified, and the subfigures A and C have been moved to Figure S7

Figure 6

(A): Heatmap of copy number variations (CNVs) in patient 8 (B-C): Enrichment of cell type(B) and tumor score(C). Normalized enrichment of different cell types and tumor score (blue and black) in patient 8 (left). Hematoxylin and Eosin (H&E) staining image (right).

Figure S7

(A ): Heatmap of copy number variations (CNVs) in patient 4 (B): Heatmap of CNVs across all the patients

5. Authors must add the number of replicates (N) in the figures' legends

Response: Thanks for reminding us. Here is the revised version of the legends for Figure 5:

(A): NCL overexpression activates the AKT pathway (B): Effect of NCL overexpression on SKOV3 cell proliferation (N = 3). SKOV3 cells were transfected with an NCL overexpression plasmid or the corresponding empty vector. Cell Counting Kit-8 (CCK8) assays were utilized for evaluating the proliferation of cells in each group at different time points. Results are presented as mean ± standard deviation (SD) from three independent experiments. ** indicates P < 0.01. (C - F): Effect of NCL overexpression on SKOV3 cell proliferation (N = 3). SKOV3 cells were transfected with an NCL overexpression plasmid or the corresponding empty vector. The gene expressions of NCL (C), phosphatase and tensin homolog (PTEN) (D), AKT1 (E), and Ki-67 (F) in each group of cells were examined using quantitative polymerase chain reaction (qPCR). Results represent the mean ± SD from three independent experiments. ** indicates P < 0.01.

6. Figure 5: to state that "NCL overexpression activates the ATK pathway", authors must test the phospho-AKT expression by western blot analysis since the only increase of AKT mRNA does not mean that it is also phosphorylated (the active form of AKT).

Response: That’s a very good perspective! Thank you for your insight. Based on your suggestion, we perform WB analysis in the cells, using the same preprocessing as our previous experiment, the results show as follows, which have further confirmed our previous conclusion, the NCL overexpression activates the ATK pathway.

Methodology part:

Western blot analysis

Cells were treated according to experimental groups, and total protein was extracted using RIPA buffer containing PMSF after cell lysis. The protein concentration was measured using the BCA protein assay. Equal amounts of protein were loaded onto SDS-PAGE gels (10% separation gel and 5% stacking gel) and separated by electrophoresis. The proteins were then transferred to PVDF membranes using a wet transfer method. The membranes were blocked with 5% non-fat milk in TBST and incubated with primary antibodies (e.g., NCL, PTEN, AKT, p-AKT, Ki67, GAPDH) overnight at 4°C. After washing, the membranes were incubated with HRP-conjugated secondary antibodies and developed using chemiluminescent substrates. Protein bands were visualized using a Tanon 5200 imaging system. Relative protein expression levels were quantified by analyzing band intensity using Image Pro Plus software. Statistical analysis was performed using one-way ANOVA, and significance was determined at P < 0.05.

Result:

Previous experiments have suggested that the suppression of NCL could lead to the upregulation of phosphatase and tensin homolog (PTEN) and reduce the activation of AKT in breast cancer[35]. Hence, we posit that NCL overexpression may similarly promote tumor cell proliferation in HGSOC (Figure 5A). To test this hypothesis, we performed experiments using an HGSOC tumor cell line SKOV3. We generated a SKOV3 cell line with NCL overexpression (Figure 5C). Then we evaluated the effect of NCL overexpression on the AKT pathway using qPCR and Western Blot. Our findings revealed a significant upregulation of AKT in gene expression level, particularly at the protein level, the expression of pAKT is significantly higher in the NCL overexpression group, while AKT protein does not have a significant difference (Figures 5E, G, J, K). We also observed significant downregulation of PTEN in both gene expression and protein levels (Figures 5D, G, I). These results indicate that NCL overexpression may promote the activation of the AKT pathway. The gene expression and protein level of Ki-67 (Figure 5F, L) and the cell proliferation assay (Figure 5B) further confirmed this finding. We found that the expression of Ki-67 in both gene and protein and cell proliferation was significantly higher in the overexpression group compared to the control. In summary, our analysis identifies the high enrichment of MDK-NCL in HGSOC tumor cell communication, and subsequent experiments validate that the overexpression of NCL indeed promotes HGSOC tumor cell proliferation through the activation of the AKT pathway.

Figure

Figure 5

(A): NCL overexpression activates the AKT pathway (B): Effect of NCL overexpression on SKOV3 cell proliferation (N = 3). SKOV3 cells were transfected with an NCL overexpression plasmid or the corresponding empty vector. Cell Counting Kit-8 (CCK8) assays were utilized to evaluate cells' proliferation in each group at different time points. Results are presented as mean ± standard deviation (SD) from three independent experiments. ** indicates P < 0.01. (C - F): Effect of NCL overexpression on gene expression(N = 3). The gene expressions of NCL (C), phosphatase and tensin homolog (PTEN) (D), AKT1 (E), and Ki-67 (F) in each group of cells were examined using quantitative polymerase chain reaction (qPCR). Results represent the mean ± SD from three independent experiments. ** indicates P < 0.01. (G - L): The effect of NCL overexpression on protein level. The protein expression levels of NCL (G, H), PTEN (G, I), AKT (G, J), p-AKT (S473) (G, K), and Ki67 (G, L) in the cells from each group were detected by Western Blot. Results are presented as mean ± SD from three independent experiments. **P < 0.01.

Supplement

Figure S9:

the original image of western blot(N=3) of AKT(A), GADPH(B), Ki67(C), NCL(D), pAKT(s473) (E) and PTEN(F)

Supplement S5 Table S4 Experiment result of qPCR , CCK8 and Western Blot

7. Abbreviations must be written in full length when mentioned for the first time

Response: Thanks for your advice, here is the revised version of all the legends:

Figure 1

(A): Workflow of the study.(B): Uniform Manifold Approximation and Projection (UMAP) plot of clustering results of spatial transcriptomics (ST) data. (C): Cluster distributions in each patient. (D): UMAP plot of marker expression for tumor (blue) and stromal (pink) samples. (E): Heatmap of pathway activity in tumor and stromal clusters.

Figure 2

(A): Spatial distribution of clusters in patients (From left to right: patient 1, patient 4, patient 8). (B): Visualization of clusters’ gene expression through dimensional reduction. (C): Heatmap of pathway activity in tumor clusters. (D): Heatmap of pathway activity in stromal clusters. (E): Heatmap of cell enrichment in two groups (divided by the different enrichment of Gene Set Variation Analysis (GSVA) signaling pathways) of stromal clusters.

Figure 3

(A - B): Clonal evolutionary tree (left) and spatial visualization (middle and right) of patient 4 (A) and patient 8 (B). (C): Heatmap of pathway activity of different clones in patient 8. (D): Changes in gene expression potentially affected by copy number variations (CNVs) in patient 8.

Figure 5

(A): NCL overexpression activates the AKT pathway (B): Effect of NCL overexpression on SKOV3 cell proliferation (N = 3). SKOV3 cells were transfected with an NCL overexpression plasmid or the corresponding empty vector. Cell Counting Kit-8 (CCK8) assays were utilized to evaluate cells' proliferation in each group at different time points. Results are presented as mean ± standard deviation (SD) from three independent experiments. ** indicates P < 0.01. (C - F): Effect of NCL overexpression on gene expression(N = 3). The gene expressions of NCL (C), phosphatase and tensin homolog (PTEN) (D), AKT1 (E), and Ki-67 (F) in each group of cells were examined using quantitative polymerase chain reaction (qPCR). Results represent the mean ± SD from three independent experiments. ** indicates P < 0.01. (G - L): The effect of NCL overexpression on protein level. The protein expression levels of NCL (G, H), PTEN (G, I), AKT (G, J), p-AKT (S473) (G, K), and Ki67 (G, L) in the cells from each group were detected by Western Blot. Results are presented as mean ± SD from three independent experiments. **P < 0.01.

Figure 6

(A): Heatmap of copy number variations (CNVs) in patient 8 (B-C): Enrichment of cell type(B) and tumor score(C). Normalized enrichment of different cell types and tumor score (blue and black) in patient 8 (left). Hematoxylin and Eosin (H&E) staining image (right). 

Reviewer #2:

Reviewer #2: This work aimed to provide a comprehensive perspective of tumor heterogeneity across the range of gene expression, copy nu

---

## [Decision Letter · Decision Letter 1]

22 Dec 2024

Deciphering Ovarian Cancer Heterogeneity through Spatial Transcriptomics, Single-cell Profiling, and Copy Number Variations

PONE-D-24-32439R1

Dear Dr. wang,

We’re pleased to inform you that your manuscript has been judged scientifically suitable for publication and will be formally accepted for publication once it meets all outstanding technical requirements.

Kind regards,

Amr Ahmed El-Arabey

Academic Editor

PLOS ONE

Additional Editor Comments (optional):

Reviewers' comments:

Reviewer's Responses to Questions

**Comments to the Author**

1. If the authors have adequately addressed your comments raised in a previous round of review and you feel that this manuscript is now acceptable for publication, you may indicate that here to bypass the “Comments to the Author” section, enter your conflict of interest statement in the “Confidential to Editor” section, and submit your "Accept" recommendation.

Reviewer #2: All comments have been addressed

2. Is the manuscript technically sound, and do the data support the conclusions?

Reviewer #2: Yes

3. Has the statistical analysis been performed appropriately and rigorously? 

Reviewer #2: (No Response)

4. Have the authors made all data underlying the findings in their manuscript fully available?

Reviewer #2: Yes

5. Is the manuscript presented in an intelligible fashion and written in standard English?

Reviewer #2: Yes

6. Review Comments to the Author

Reviewer #2: Deciphering Ovarian Cancer Heterogeneity through Spatial Transcriptomics, Single-cell Profiling, and Copy Number Variations Accept

7. PLOS authors have the option to publish the peer review history of their article (what does this mean? ). If published, this will include your full peer review and any attached files.

**Do you want your identity to be public for this peer review?** For information about this choice, including consent withdrawal, please see our Privacy Policy .

Reviewer #2: **Yes: ** mohnad abdalla

---

## [Editor Report · Acceptance letter]

PONE-D-24-32439R1

PLOS ONE

Dear Dr. Wang,

I'm pleased to inform you that your manuscript has been deemed suitable for publication in PLOS ONE. Congratulations! Your manuscript is now being handed over to our production team.

Kind regards,

on behalf of

Dr. Amr Ahmed El-Arabey

Academic Editor

PLOS ONE